# Ordovician–Silurian true polar wander as a mechanism for severe glaciation and mass extinction

Xianqing Jing[1], Zhenyu Yang [1 ✉], Ross N. Mitchell [2 ✉], Yabo Tong[3], Min Zhu [4] & Bo Wan [2]

The Ordovician–Silurian transition experienced severe, but enigmatic, glaciation, as well as a paradoxical combination of mass extinction and species origination. Here we report a large and fast true polar wander (TPW) event that occurred 450–440 million years ago based on palaeomagnetic data from South China and compiled reliable palaeopoles from all major continents. Collectively, a ~50° wholesale rotation with maximum continental speeds of ~55 cm yr$^{-1}$ is demonstrated. Multiple isolated continents moving rapidly, synchronously, and unidirectionally is less consistent with and plausible for relative plate motions than TPW. Palaeogeographic reconstructions constrained by TPW controlling for palaeolongitude explain the timing and migration of glacial centers across Gondwana, as well as the protracted end-Ordovician mass extinction. The global quadrature pattern of latitude change during TPW further explains why the extinction was accompanied by elevated levels of origination as some continents migrated into or remained in the amenable tropics.

The Earth system underwent critical changes during the Ordovician–Silurian (O–S) transition 460–435 million years (Ma) ago. The end-Ordovician mass extinction, which can be regarded as the second most lethal of the "Big Five" mass extinctions, replaced much of the Cambrian marine fauna with later Paleozoic fauna[1]. Accompanying the O–S mass extinction was the first of three occurrences of significant glaciation during the Phanerozoic Eon (ca. 541 Ma to present), of which only ~25% represented glacial intervals[2]. The O–S glacial interval is unusual both because it was short-lived and occurred at a higher atmospheric partial pressure of $CO_2$, perhaps 8–16 times higher than today[3]. There are multiple hypothesized causes for the Ordovician extinction, including intense volcanic eruptions and/or large igneous provinces[4–7], oceanic anoxia[4,6], special paleogeography[8], large and short-lived glaciation[8–10], and even the evolution of land plants[9,10]. The diversity of proposed mechanisms thus reflects the myriad changes in the atmosphere, biosphere, hydrosphere, lithosphere, as well as in the mantle at that time.

Among the candidate mechanisms behind the widespread O–S global change, intense volcanism and paleogeography are generally regarded as the basic causes for the other changes[4,6–8]. However, there is still debate over how exactly volcanism impacted the environment[4], with some arguing that it resulted in global warming, while others claiming it caused glaciation. The volcanism theory has also been used to explain the traditional two-pulse extinction model[5], but recently reported high-resolution biodiversity curves[1,7,11] suggest instead a protracted extinction rather than the simple traditional two-pulse model. Therefore, the mechanisms once fit to a two-pulse extinction model may no longer be applicable, or at least require modification.

Paleogeography is another critical boundary condition for understanding such marked transitions in Earth's surface environment,

[1]College of Resources, Environment and Tourism, Capital Normal University, Beijing, China. [2]State Key Laboratory of Lithospheric Evolution, Institute of Geology and Geophysics, Chinese Academy of Sciences, Beijing, China. [3]Institute of Geomechanics, Chinese Academy of Geological Sciences, Beijing, China. [4]Key Laboratory of Vertebrate Evolution and Human Origins of Chinese Academy of Sciences, Institute of Vertebrate Paleontology and Paleoanthropology, Chinese Academy of Sciences, Beijing, China. ✉e-mail: zhenyu.yang@cnu.edu.cn; ross.mitchell@mail.iggcas.ac.cn

but the prevalent palaeogeographic models used[5,6,8,12] are imprecise, lacking palaeolongitude control and high temporal resolution. Constraining palaeolongitude is particularly important when continents are dispersed as they were in the early Paleozoic during the transition between supercontinents Rodinia and Pangaea. Temporal resolution is critical when continents are moving fast and multiple kinematic models suggest some of the highest continental motions of the Phanerozoic Eon occurred during this time[13]. It is therefore difficult to evaluate the exact impact palaeogeographic changes may have had on the end-Ordovician environmental changes. For example, employing prevalent palaeogeographic models, biogeochemical models[14] fail to both recreate the environmental changes during the critical Hirnantian stage as well as to explain the migration of the glacial centers[15,16].

Similar extreme transitions in Earth's surface conditions occurred during the preceding Ediacaran–Cambrian periods, and this interval has been proposed to have experienced large-scale (60–90°) true polar wander[17–23]. True polar wander (TPW) is the movement of the entire solid Earth (mantle and crust) relative to Earth's spin axis in order to stabilize Earth rotation. It is different from the plate motion of plate tectonic theory, which proposes that tectonic plates, including continents or not, move over the asthenosphere relative to the underlying convecting mantle. Tectonic plates move in different directions and with different velocities (even in the case symmetric seafloor spreading, the directions of motion are opposite of each other). In contrast, TPW can induce wholesale polar motion of the plates unidirectionally and synchronously, thus changing paleogeography rapidly and globally. Therefore, TPW potentially impacts much of Earth system evolution including changing ocean currents, air circulation, relative sea level, and depocenters of the carbon cycle[18,21,24–26].

Van der Voo[27] first proposed a round-trip TPW oscillation (two sequential back-and-forth TPW events) during the Late Ordovician to Late Devonian based on sparse and roughly-dated palaeomagnetic poles from three continents exhibiting similar large and rapid segments of apparent polar wander (APW)[27]. The proposed ~40° amplitude of the putative O–S TPW event (the first event of the pair in the oscillation), if shown to be valid, would represent the largest TPW event in the past 500 million years[28]. Although Piper et al.[29] revisited this TPW interval, they only studied the late Silurian–Early Devonian part. No follow-up research has reexamined the Ordovician–Silurian TPW event, which, as proposed, is too crude to assess its validity nor its potential impact on the environmental and ecological changes occurring at that time. Many new palaeomagnetic results during this purported interval of large-scale TPW have been reported since[30–34], justifying a reexamination. Furthermore, although putative O–S TPW was proposed[27] prior to most of the numerous hypotheses attempting to explain the variegated aspects of global change during this transition, O–S TPW has never before been taken into account for its potential environmental effects. Even for those models involving palaeogeography as a critical aspect for the changing surface conditions during this dramatic environmental transition, TPW has been neglected.

In this work, we present palaeomagnetic data from South China as well as compile data globally from 6 continents to provide a rigorous and high-resolution palaeogeographic reconstruction of the O–S transition. Our results demonstrate the occurrence of a large and rapid TPW rotation synchronous with the environmental changes across the O–S boundary. The heretofore enigmatic features of global change during this time interval can collectively be reconciled by this refined TPW-based palaeogeographic model, explaining both glacial and extinction dynamics.

## Results

### New palaeomagnetic data from South China

Previous palaeomagnetic data from South China tentatively suggest there may have been a rapid continental movement during the Late Ordovician to early Silurian[30,32,35]. However, data from the Silurian have been calculated as a mean pole for the whole Period (443.8–419 Ma)[32,35] (Supplementary Fig. 1), which precludes detailed evaluation of maximum rates of continental motion during the O–S transition. Due to its importance for palaeogeographic comparison before and after the O–S boundary, the upper Telychian strata of the Huixingshao Formation (ca. 436–435 Ma) in Xiushan county, Chongqing, South China (Supplementary Figs. 1 and 2) were selected for detailed palaeomagnetic study. Standard palaeomagnetic methods were employed and are detailed in the Methods. Stepwise thermal demagnetization revealed a stable component with high unblocking temperature suggestive of a remanence carried by hematite, which is also supported by rock magnetic experiments (Fig. 1 and Supplementary Figs. 3, 4, and 6). Detailed description of the palaeomagnetic results is provided in the Supplementary information. The magnetostratigraphic record reveals at least four coherent polarity zones (Fig. 2) strongly suggesting that the high-temperature component from section Yongdong (SY) is primary and can be used for palaeogeographic reconstructions. However, the $K$-value of dispersion of the virtual geomagnetic poles (VGPs) of these six sites is 90.3 (Supplementary Table 1), exceeding 70, which suggests that these data may not average out the palaeosecular variation (PSV)[36]. To overcome this issue, we sought to combine our new data with the most reliable coeval previous data.

We reassign the ages of existing Silurian palaeomagnetic results[32,35] according to a recently updated stratigraphic timescale[37–39] (Supplementary Fig. 1). A notable revision in these age reassignments is that the Rongxi Formation previously regarded as ca. 420 Ma in age is in fact early Telychian (ca. 438.5–437 Ma) (Supplementary Fig. 1). Again, data from these previous studies[32,35] seem not to average out PSV[36] (Supplementary Table 1; detailed analysis in Supplementary information). Nonetheless, after combining all data from the Rongxi and Huixingshao Formations (total 28 sites), a $K$-value of 48.4 is achieved, which is below 70 and suggests sufficient averaging of PSV. Furthermore, these data also pass a fold test[40] at 99% confidence (k in geographic coordinates is 7.64, in stratigraphic coordinates is 31.17). This new early Silurian pole (S1M) calculated by averaging the VGPs from the Rongxi and Huixingshao Formations plots far from all younger poles and earns a reliability index of 6 of 7 (ref. 36, Supplementary information). Intriguingly, the new early Silurian pole (S1M) plots far from (≥50°) away a high-quality Late Ordovician (late Sandbian–middle Katian; 454–448 Ma, or ca. 451 Ma) pole of South China[30] (Figs. 1l, 3a).

Given only ~10 Ma between these two ages, the 54.4° ± 6.4° arc distance between these two poles indicates a rapid APW rate of 5.4 ± 0.6 Ma⁻¹ for South China. During this time interval, South China experienced a region tectonic movement, however it was restricted to only its southeastern part (Cathaysia terrane)[41]. Our early Silurian data and the Late Ordovician data are from northwestern South China (upper Yangtze terrane), which was largely unaffected by this tectonism. In addition, the regional tectonism should have only induced large differences in the declination of these data (due to potential vertical-axis rotation), but cannot explain the large inclination difference that is observed corresponding to a ~28.5° change in palaeolatitude. Non-uniformitarian magnetic fields (e.g., quadrupolar or octupolar) may also result in apparent changes in latitude[33]. However, in order to explain the reduced inclination of the Late Ordovician data (35°) to our Silurian data (18°), one would have to claim a same-sign octupole that was stronger than 20%, which is more extreme than any previous claims in the Phanerozoic[42], and an opposite-sign octupole would increase, not decrease, inclination. Furthermore, both non-dipole cases would only affect inclination and therefore cannot explain the even larger anomaly in terms of the ~59° declination change. Lastly, an oscillation between polar and equatorial dipoles (if possible on Earth) could affect declination[43],

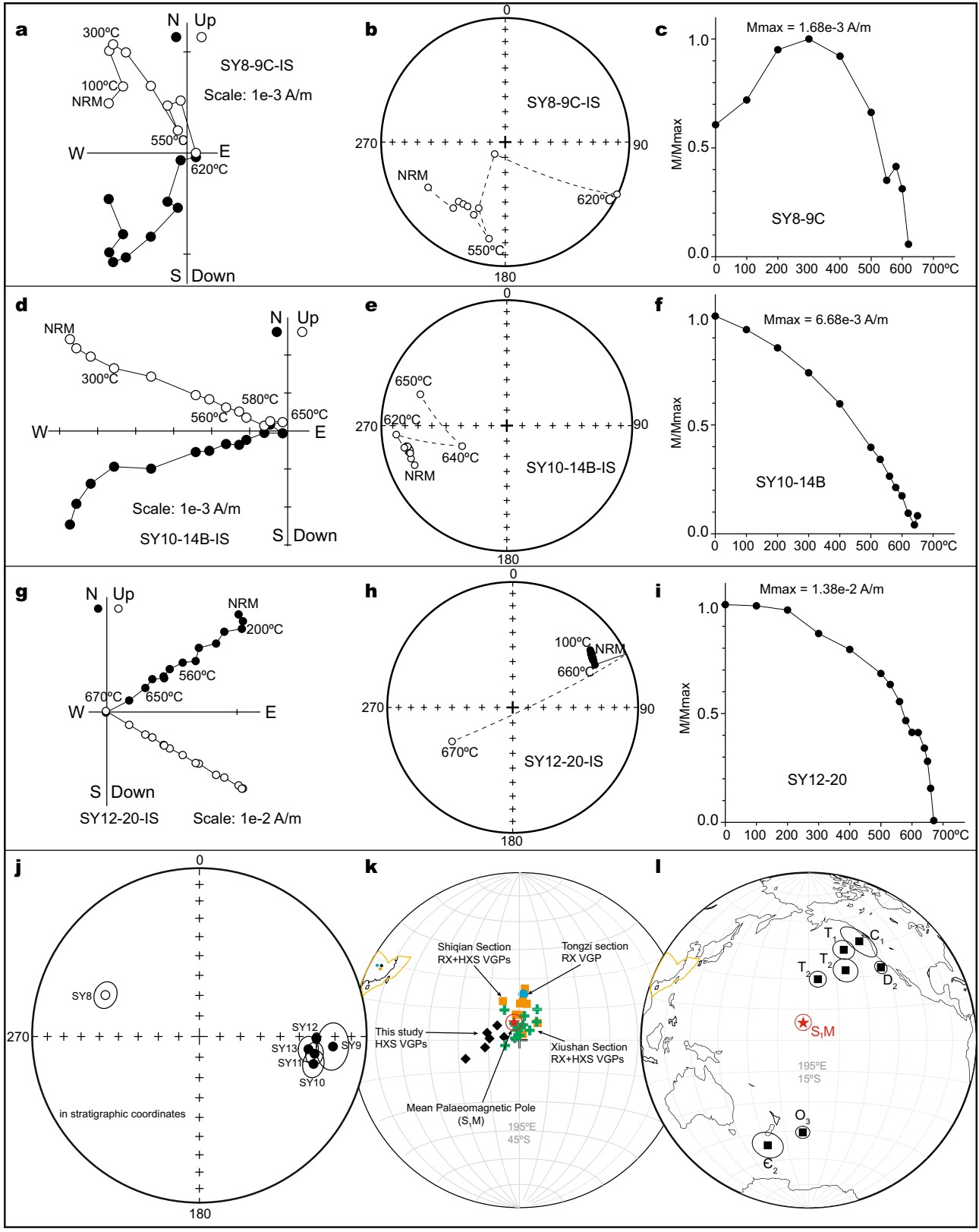

but would predict a -90˚ change that is not observed. Therefore, we argue that this large and rapid motion of South China corroborates from an additional continent the proposed O–S TPW event[27], albeit with an even larger amplitude than once thought. Nevertheless, any reproducibility test of TPW should aim to be global in scope, so we must consider the palaeomagnetic records of the other major continents.

## Late Ordovician-early Silurian true polar wander

Strikingly, in addition to the large-scale 54° ± 6° APW of South China, the Late Ordovician–early Silurian palaeopoles from Tarim, Siberia, Baltica, and Gondwana also all demonstrate large arc distances of APW: 54° ± 9°, 47° ± 17°, 55° ± 14°, and 58° ± 21°, respectively (Fig. 3a, Supplementary Table 2), with associated APW rates of 5.4° ± 0.9°, 4.7° ± 1.7°, 5.5° ± 1.4°, and 5.8° ± 2.1° Ma⁻¹, respectively. Data from

**Fig. 1 | New Silurian palaeomagnetic data from South China and compilation with previous results.** Zijderveld plots (**a**, **d**, **g**), equal area projections (**b**, **e**, **h**) and normalized stepwise thermal decay curves (**c**, **f**, **i**) of the thermal demagnetization of representative samples from the section at Yongdong (SY) in geographic coordinates. In the Zijderveld plots, black and white dots represent horizontal and vertical projections, respectively, while in the equal area projections, they represent directions plotted in the lower and upper hemispheres respectively. **j** Equal area stereographic projection of site mean directions of the high-temperature components of the Huixingshao (HXS) Fm from this study in stratigraphic coordinates. **k** Virtual geomagnetic poles (VGPs) of HXS Fm from the SY section from this study compared with VGPs from the HXS Fm and the Rongxi (RX) Fm from Opdyke et al.[35] and Huang et al.[32]. Resulting combined early Silurian pole (S₁M) using all data from the RX and HXS Formations from this study and previous work is shown as red star with associated cone of 95% confidence. **l** The new recalculated early Silurian pole (S₁M) is distinct from existing poles of South China[71]. All plots were generated with PaleoMac[72].

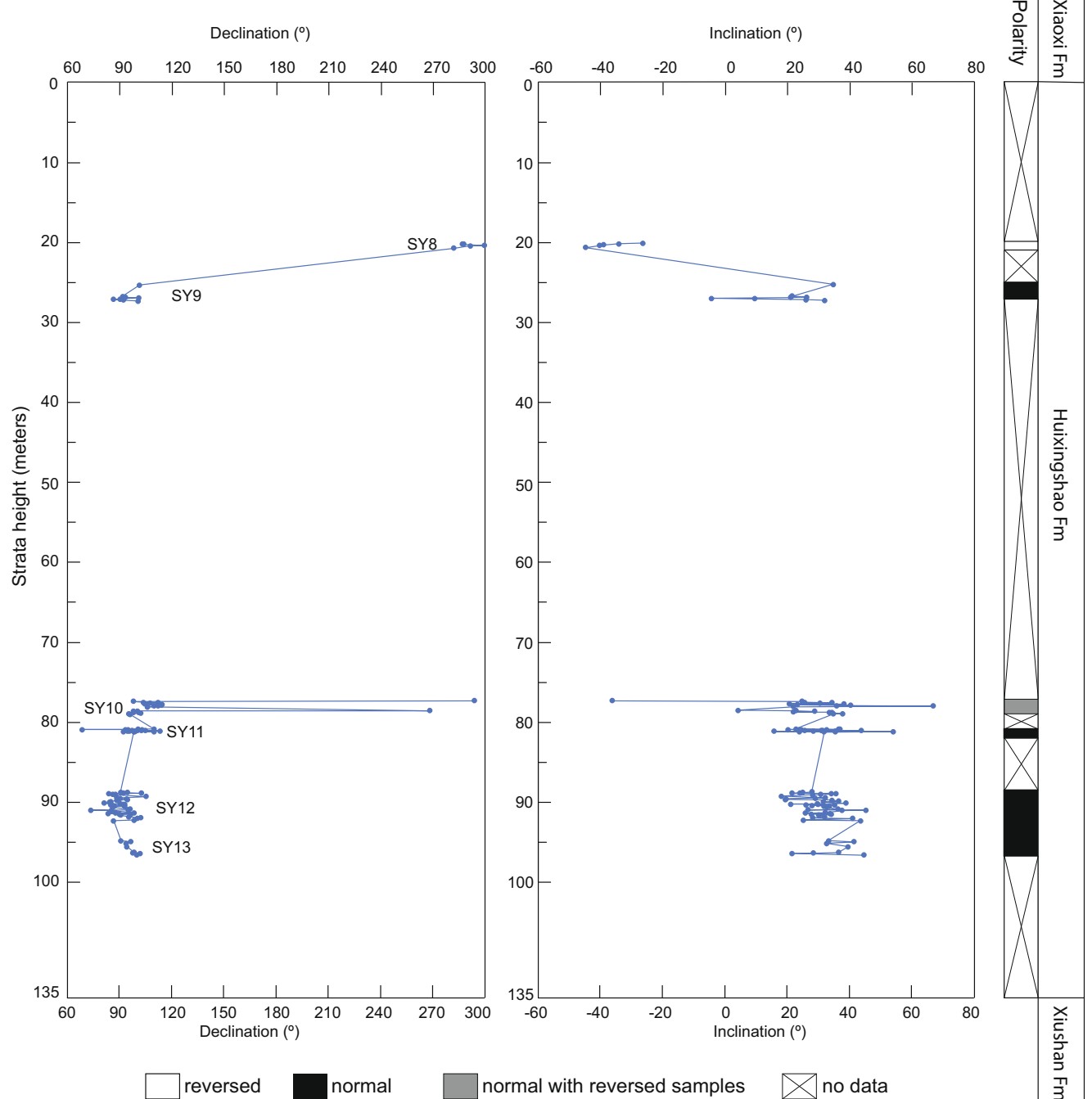

**Fig. 2 | Magnetostratigraphy of the Huishingxiao Formation.** Sampled section at Yongdong (SY). Directions with declinations >240° were interpreted as reversed polarity, and otherwise, as normal polarity.

Baltica and Gondwana represent recent synthetic APW paths, which consider the age error and the quality of the data[34]. For comparison, we also calculate the arc distances for Baltica and Gondwana from 450–430 Ma using the synthetic APW paths of Torsvik et al.[44] (Supplementary Table 3), which are 51.2° ± 8.2° and 24.5° ± 18°, respectively. While the results for Baltica agree with both methods, the large difference of the two synthetic APW paths for Gondwana reflect either the larger 20 Ma age bins of Torsvik et al.[44] oversmoothing the

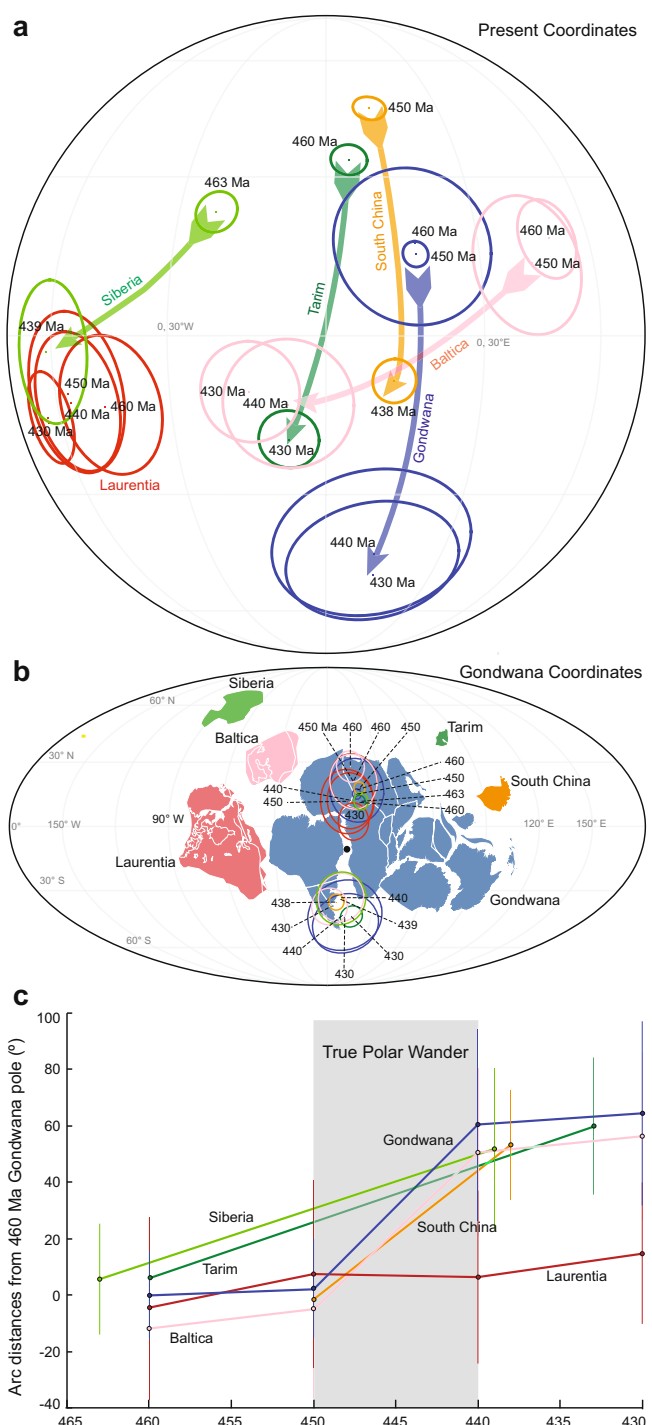

**Fig. 3 | Ordovician–Silurian apparent polar wander paths globally. a** Global palaeomagnetic poles for 460–430 Ma from South China, Tarim, Siberia, Baltica, Laurentia, and Gondwana. Apparent polar wander (APW) paths all exhibit (except Laurentia) large shifts between ca. 450–440 Ma. Poles shown in present-day coordinates. Pole information is listed in Supplementary Table 2. **b** Poles rotated into the common reference frame of Gondwana illustrating the APW overlap. Black dot is a reference point in central Gondwana used for palaeolatitude estimates in Fig. 6e. **c** APW arc distances for all poles globally (in Gondwana reference frame of **b**) relative to the 460 Ma Gondwana pole (as an arbitrary reference point before the hypothesized true polar wander event). See text for discussion of Laurentia. Vertical bars are intervals of 95% confidence. Plots in **a** and **b** were generated with GPlates[73].

data and/or the lack of poles during this time interval which is non-ideal for synthetic methods. Nonetheless, at least four continents demonstrate similar large amplitude and synchronous polar motion. As discussed, regional tectonics and non-uniformitarian geomagnetic fields cannot explain this systematic global APW anomaly. Plate motion, driven by slab subduction and mantle convection, also cannot explain these synchronous and similar large amplitude movements of multiple isolated continents either, as it requires relative motion between different plates with different velocities (speeds and/or directions).

TPW could explain the large and synchronous dispersions of O–S palaeopoles globally. TPW is rate-limited by the ability of the viscous mantle to deform into a reoriented hydrostatic figure[45,46]. TPW can occur as fast as the fastest plate motion or even comparatively faster, particularly in more ancient times when the mantle was hotter, less viscous, and thus more deformable[46,47]. Numerical simulations suggest that a 40–50° amplitude TPW event can occur in ~10 Ma if the viscosity of lower mantle is $10^{22}$ Pa s[46]. Presently lower mantle viscosity is about $3 \times 10^{22}$ Pa s[48], while it may be 3 times lower at 450 Ma[47]. Hence, considering almost all continents sped up synchronously, we propose that during Late Ordovician, most likely after the middle Katian Stage but before the Silurian early Telychian Stage, a TPW event occurred. Furthermore, the fact that all the ~50° arc distances of APW are within statistical uncertainty of each other means that the data pass the global reproducibility test of TPW.

We note that palaeomagnetic poles from Laurentia during this time are characterized, in contrast, by much less APW, and almost essentially a stillstand (Fig. 3). At face value, one continent with a statistically different arc distance of APW compared with those of other continents does not invalidate the TPW hypothesis[49]. This point of caution is particularly relevant here because during this time Laurentia was an isolated plate with its own tectonic motion vector. In the Paleozoic, the Iapetus and Rheic oceans that existed in between Laurentia and West Gondwana rapidly expanded and vanished[12,34], which certainly would have resulted in fast tectonic movements of Laurentia and may seem at odds with its small amount of APW. As the tectonic motion of Laurentia during the closure of the Iapetus would have been mostly opposite to its sense of motion due to TPW, the effect of TPW would be partially offset and thus should appear as a relative stillstand, where APW = plate motion + TPW. In this sense, as Laurentia would have undergone large tectonic motion during this time, its palaeomagnetic stillstand can only be reconciled if TPW in the opposite direction is invoked. Thus, the TPW event inferred from all other continents provides a convenient way to explain the prior paradox of a Laurentian APW stillstand during the closure of the Iapetus Ocean.

We should also note that, strictly speaking, Laurentia may not exhibit a total stillstand. The circles of 95% confidence of the youngest and oldest Laurentian poles (460 and 430 Ma) only very slightly overlap, and the results of an F test[50] demonstrate that the poles are distinct from each other at the 99% confidence level ($F = 10.6$). This test indicates that the $18.9° \pm 19.3°$ arc distance between the two O–S poles is statistically significant. Therefore, while the presumably considerable tectonic motion of Laurentia partially masks the ~50° TPW event, Laurentia nonetheless does indeed record a statistically significant portion of the TPW amplitude that, in reconstructed coordinates, is consistent with the sense of TPW rotation more clearly recorded on the other continents. Otherwise, this relative stillstand may be an artifact of the large age errors of these poles used for APW comparison[34].

As defined as the migration of the maximum moment of inertia ($I_{max}$) to align with Earth's spin axis, TPW occurs as a rotation about an Euler pole controlled by the minimum moment of inertia ($I_{min}$) that is

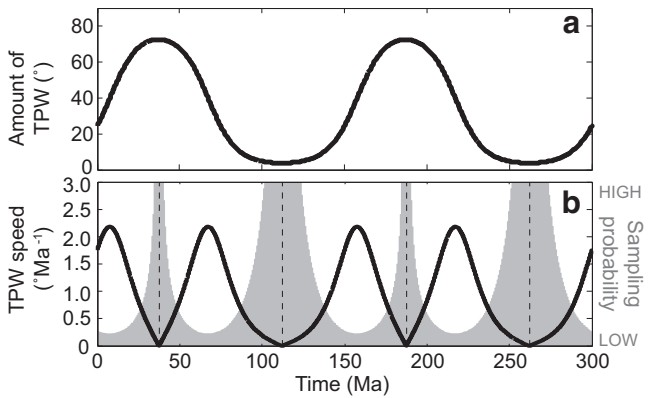

**Fig. 4 | Probability of sampling true polar wander. a** True polar wander (TPW) angle as a function of time with an initial condition of 25°. **b** TPW speed (black line) and probability function (shaded gray) as a function of time.

equatorial and is therefore predicted to circumscribe a great-circle APW path. Identifying TPW as a great-circle APW path also assumes that plate motion of the continent relative to the mantle is negligible, the change in the orientation of the principal axes of non-hydrostatic moment of inertia is instantaneous, and those subsequently do not change at all. The similar amplitude and synchronicity of these five continents indicate their individual plate motions are negligible relative to the shared TPW motion. Numerical simulations indicate such a change in the orientation of the principal axes of non-hydrostatic moment of inertia can be completed within 10 Ma[46]. There is also a notable absence of poles in between the before/after poles recording the TPW shift (Fig. 3). These systematic gaps in the APW paths of all continents are consistent with the stroboscopic effect expected for TPW, which is a non-linear process that speeds up and slows down, thus rendering it less likely for rocks to form (making them available for palaeomagnetic sampling) during the peak rate of TPW in the middle of the event. A simple simulation (Methods) demonstrates that it is 20 times less likely to sample TPW "in action" than the endpoints largely before/after the TPW event (Fig. 4). This inherent bias can explain why the O–S TPW event is sampled exclusively by endpoints for all continents. We therefore confirm and refine the original proposal[27] of a large amplitude ~50° TPW event occurring across the O–S boundary.

Given this was a time of major plate tectonic reorganization in between assembly of megacontinent Gondwana and its larger supercontinent Pangaea[51], there is no shortage of potential sources of subduction-related mass anomalies that might have provided the excitation for the large-scale TPW event across the O–S boundary. The Australian Tasmanides, the Laurentian Appalachians, and the Baltic Caledonides were all active at this time; however, provided their positions relatively close the TPW axis ($I_{min}$), their influence on Earth's rotation would have been dampened compared to mass anomalies elsewhere. In contrast, both the Proto-Tethyan and Terra Australis subduction systems on either side of Gondwana were ~90° away from $I_{min}$ and thus in the plane of TPW containing $I_{max}$ and $I_{int}$ would have been ideally positioned relative to Earth's prolate non-hydrostatic figure to have excited large-scale TPW.

In the Late Ordovician, the Proto-Tethyan system experienced a fundamental shift from subduction to collision[52]. Both the timing (pre-TPW) and the sense of this change in slab dynamics—with the foundering oceanic slab likely ponding at the mantle transition zone, thus causing a positive anomaly in the geoid kernel driving TPW for this region equatorward[53]—are consistent with the observed palaeogeographic shift of the Tethyan subduction zone from mid-latitudes into the tropics (Fig. 5). Also, in the Terra Australis system on the other side of Gondwana, an intriguing coincidence is that the new position of the

South Pole (post-TPW) becomes centered on the Antarctica–South America segment of the subduction system (Fig. 5) that experienced a dramatic shift from negative to positive hafnium isotopes at this age[54]. Such a shift due to increased mantle-derived magmatism in the arc indicates slab retreat, which can occur before slab break-off as a slab meets resistance to subduction after impinging the mantle transition zone[55]. Because of the time lag between slab subduction in the upper mantle and its penetration into the lower mantle, a dramatic slab avalanche from the upper into the lower mantle after stagnation at the mantle transition zone could thus conveniently explain the new pole position assumed in the Silurian as the geodynamic change in the Terra Australis would have driven TPW for this region poleward[53]. Thus, the dramatic changes in slab dynamics of both subduction systems on either side of Gondwana could have contributed to the collective forcing behind the largest TPW event in the past 500 million years.

It is also possible that the waxing and waning of ice sheets across Gondwana contributed to the mass anomalies driving O–S TPW, or there was some feedback between TPW and glaciation. In particular, there is a migration of glacial centers from northern Africa to southern Africa–South America, where glacial and periglacial strata in the former region are predominantly Ordovician and those in the latter neighboring regions are predominantly latest Ordovician or Silurian[15]. That is, the mass load associated with incipient Ordovician glaciation applied in northern Africa could have been driven to the equator by TPW, causing southern Africa–South America to move to the pole and thus moving the glacial center there in the earliest Silurian (Fig. 5). This hypothesis, by extension, would also predict ensuing oscillatory Silurian–Devonian TPW back in the direction of northern Africa (in order to drive the glacial center in southern Africa–South America equatorward), which has indeed been previously hypothesized[27], but the assessment of which is beyond the scope of our study on O–S TPW. In the Cenozoic, however, glaciation is typically regarded more as an effect of TPW rather than a cause of it[56], as the amplitude of glacially induced TPW is smaller than TPW driven by mass reorganizations in the mantle[56]. Nevertheless, given the larger size of the Paleozoic pan-Gondwanan ice sheet, and thus its presumably larger mass load, glacial loading deserves further investigation for potentially driving the O–S TPW event. If valid, such an interpretation—the incipient glacial load causing TPW, which then led to more severe glaciation as Gondwana became centered over the South Pole—presents a fascinating potential feedback between TPW and glaciation.

### Palaeogeographic reconstructions based on true polar wander

Traditionally, the superposition of APW paths is used to reconstruct the configuration of different continents during time intervals of supercontinentality[57]. However, during times of plate tectonic reorganization in between supercontinents, this method cannot be used to reconstruct isolated continents that are in relative motion, which is most likely how the end-Ordovician world was kinematically configured[12]. Nonetheless, when APW is predominantly driven not by plate motion but by TPW, then the superposition of APW paths can be used to determine the relative positions of different continents whether they are united or isolated because the TPW motion is shared by all continents and thus provides a common global reference frame[26,58]. Such an APW comparison only requires a minimum of two poles from before and after the TPW event. Therefore, we can accurately reconstruct global paleogeography of the major continents across the O–S boundary by leveraging TPW.

To make our reconstructions, northwest Africa is fixed and all the other continents are rotated into northwest African coordinates (Euler rotation parameters listed in Supplementary Table 4). We first fitted a great circle to the palaeopoles from Gondwana, of which northwest Africa is a part (Fig. 5a). Poles from all the other continents were then rotated to overlap the Gondwanan poles at their corresponding ages. The TPW-based reconstructions constrain the relative positions of all

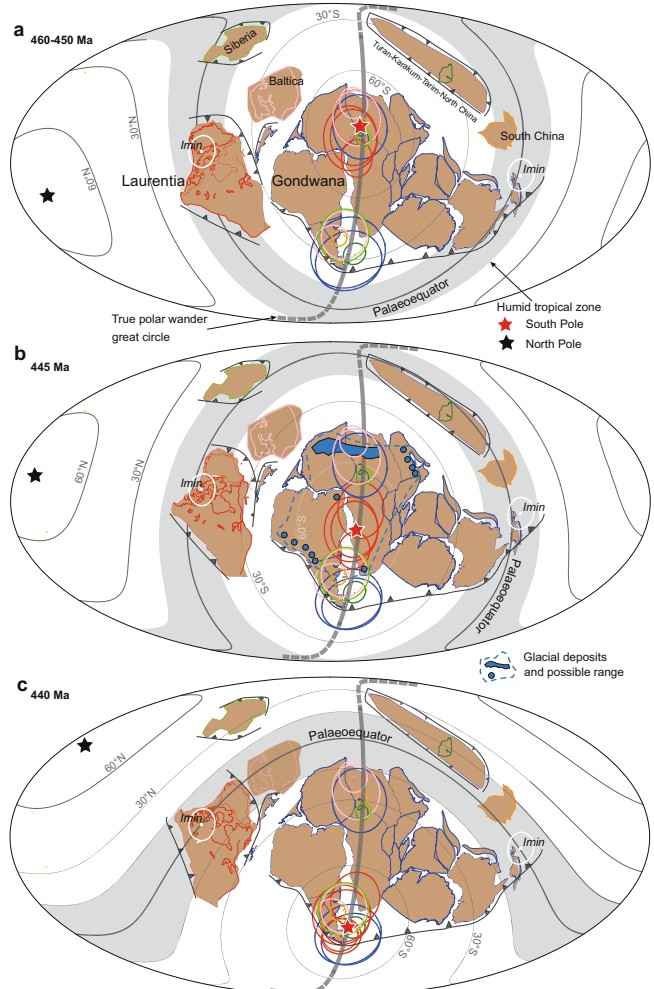

**Fig. 5 | Palaeogeographic reconstructions based on Ordovician–Silurian true polar wander.** Reconstructions for: (**a**) 460–450 Ma, (**b**) 445 Ma, and (**c**) 440 Ma. Palaeomagnetic poles are color-coded as in Fig. 3a. $I_{min}$, minimum moment of inertia (equatorial true polar wander axis of rotation). Palaeomagnetic poles (and associated continents) of each age are rotated to coincide with the South Pole, although a 5–10° range of flexibility is occasional adopted as is common praxis in ancient palaeomagnetic reconstructions. The latitudinal band between 15° north and south of the Equator is assigned as the humid tropical zone with intensive chemical weathering. Maps are shown in Mollweide projection. For better displaying their distribution, we fixed the continents and rotate the Mollweide projection to fit the Palaeo-south pole. All plots were generated with GPlates[73].

these continents during 460–440 Ma, not only including palaeolatitude constraints, but also commonly unconstrained relative palaeolongitude. As mentioned, the essentially opposite tectonic motion of Laurentia effectively cancels out some of the TPW rotation for Laurentia, therefore its position relative to other continents changes over time.

Three high-resolution, TPW-based palaeogeographic reconstructions are provided at 460–450, 445, and 440 Ma (Fig. 5). The 445 Ma reconstruction is an interpolated position between 450 and 440 Ma. Subduction zones and the evolution of Avalonia is simplified from Cocks and Torsvik[12]. A salient difference between our reconstructions and previous ones[5,6,8,12,59] is that Gondwana rapidly swept over the South Pole (Figs. 5, 6e). Meanwhile, during 460–450 Ma, the Niger–Chad zone was located at the South Pole rather than the Morocco–Algeria zone (Fig. 5a). At 460–450 Ma, Gondwana was distributed from the South Pole to the Equator, with the majority of the landmass located at high-to-mid latitudes (Fig. 5a). Laurentia straddled the equator, with its east coast (present coordinates) outside of the tropics. The positions of

Baltica and Siberia are similar to previous reconstructions[5,6,12,59]. Constrained using the APW path of Tarim, Turan–Karakum–Tarim–North China[60] is constrained to a position between South China and Siberia. Most of South China was in the tropics, which is consistent with the palaeoequatorial setting suggested by the mega-nodular limestone, a time-specific carbonate facies[61].

After 450 Ma, TPW initiated a dramatic change in palaeogeography. At 445 Ma, in the middle of the TPW event, northern Africa moved off the South Pole, where it was replaced by southern Africa and South America (Fig. 5b). (In terms of tectonic motions, Laurentia moved closer to Baltica, but farther from Gondwana because of the fast opening of the Rheic Ocean.) During the TPW event, Baltica and Avalonia moved into low latitudes, and Siberia, Turan–Karakum–Tarim–North China, and South China ended up straddling the equator and were nearly all located within the tropics (Fig. 5b). After the TPW event was over by 440 Ma (the Silurian), northern Africa and Arabia occupied low latitudes, while South America and southern Africa were located around the South Pole (Figs. 5c, 6e). Siberia and Turan–Karakum–Tarim–North China all moved out of the tropics, while Baltica moved into the tropics and South China moved into the tropics of the northern hemisphere. By the Silurian, more continents were positioned at mid-to-low latitudes (more than ~14,000,000 km²; Fig. 5c) than before (Fig. 5a, b).

## Discussion

Having verified and refined the existence of a large-scale TPW rotation across the O–S boundary and reconstructed the associated rapid continental motions more precisely than ever before, we consider the potential impacts of such an extreme palaeogeographic disruption on the variegated changes to Earth's surface conditions during this enigmatic time of transition. One unique puzzle of end-Ordovician global change is the occurrence of severe glaciation. Continental configuration is known to play an important role in setting the climate state on both long and short time scales[2,62], so any changes in paleogeography due to TPW should also be critically considered. Phanerozoic glaciations have been generally correlated with the occurrences of arc-continent collisions distributed within the humid tropical zone, which serves to lower global temperatures by increasing silicate weathering and consuming $CO_2$, an atmospheric greenhouse gas[2,62].

Reliable evidence of O–S glacial deposits from Gondwana that was positioned near the South Pole at the time is mostly found during the Hirnantian stage, with only sporadic cases reported before and after[63]. The latest Katian and early Silurian glaciogenic sediments occurring before and after the Hirnantian, respectively, found in Niger and South America argue for a medium-to-large-scale glacial interval, if using previous paleogeography[5,6,8,12,59]. However, no other sectors of Gondwana record glaciogenic sediments during these time intervals. Furthermore, cyclostratigraphic analysis also indicates the main glaciation initiated in the early Hirnantian stage[64]. In our reconstruction of the latest Katian, Niger was close to the South Pole (Fig. 5a), where glaciation is most likely to develop as it is the coldest place on Earth because incoming solar radiation is reduced by the high angle of incidence at high latitudes, as evidenced in the current polar ice caps of Antarctica and Greenland. Meanwhile, during the early Silurian, South America moved over the South Pole (Fig. 5c) and glaciation, as expected, occurred there again. Therefore, our reconstructions match the migration of glacial centers across Gondwana quite well[15,16,63], which in turn independently validates our proposal of O–S TPW. In particular, the fact that the short, sharp Hirnantian glaciation[3] is shown to have occurred precisely in the middle of the TPW event—as Gondwana swept over the South Pole (Figs. 5, 6e) and its ice sheet presumably should have expanded to its largest size for a very brief time interval—indicates that TPW is a previously unrecognized major factor in the total causal nexus explaining the Hirnantian glaciation, as for the Cenozoic northern hemisphere glaciation[56,65].

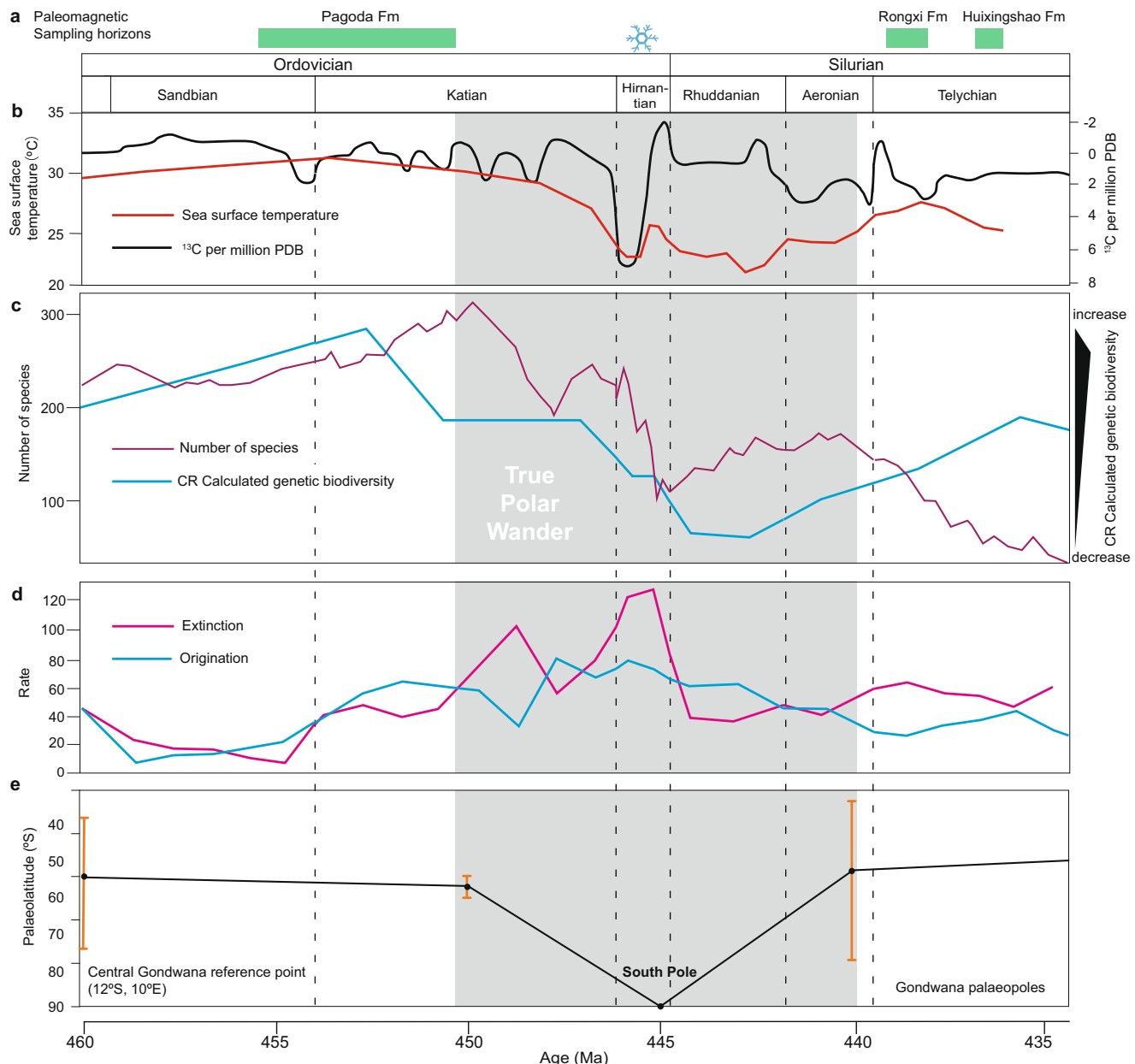

**Fig. 6 | Late Ordovician–early Silurian global change and true polar wander.**
**a** Temporal distribution of the palaeomagnetic sampling horizons by Formation (Fm) from South China. Snowflake above the Hirnantian stage indicates the age of short, sharp glacial advance[3]. **b** δ[13]C record (black) and sea surface temperature variation (red) from Rasmussen et al.[7] **c** Biodiversity during the Sanbian–Telychian stages from Deng et al.[11] (purple line) and Rasmussen et al.[7] (blue line). CR capture-recapture modeling. **d** Rates of origination (blue) and extinction (magenta) with 1 million year age binning from Deng et al.[11] **e** Palaeolatitudinal variation of a reference point in central Gondwana (12ºS, 10ºE) calculated by using 460–430 Ma Gondwana palaeopoles listed in Supplementary Table 2. Note that the 430 Ma palaeolatitude is not displayed. Orange vertical bars are intervals of 95% confidence.

We next consider how TPW interacted with other factors previously thought to control Late Ordovician early Silurian glaciation. Volcanic eruption[4,7], plant and large phytoplankton evolution[9,10], and silicate weathering[2,62,66] have all been proposed to explain the extreme climate change across the O–S boundary. However, previous palaeogeographic constraints limited the accuracy of such interpretations[6,14,59]. Our reconstructions demonstrate that after 450 Ma, TPW placed Siberia, Turan–Karakum–Tarim–North China, and South China entirely into the tropics; in addition, a large portion of Gondwana was moved down from high- to mid-latitudes (Fig. 5b). All these palaeogeographic changes thus favor the observed intensification of silicate weathering helping drive cooling. These palaeogeographic conditions resulted in not only the Hirnantian glaciation,

but also the marked positive Hirnantian carbon isotope excursion (Fig. 6b) due to the increased fraction of organic carbon burial resulting from the preponderance of tropical continental margins (Fig. 5b) analogous to the modern Amazon River. Previous studies suggest that arc-continent collisions within the humid tropical zone set Earth's climate state to first order during the Phanerozoic Eon[2]. After the peak glaciation and positive carbon isotope excursion, a large portion of the arc-continent collisions of Laurentia, Siberia, and Turan–Karakum–Tarim–North China moved out of the tropics (Fig. 5c), thereby silicate weathering plummeted causing deglaciation and carbon cycle recovery (Fig. 6).

Although there are myriad ways in which TPW may indirectly affect biodiversity through environmental change[26], a direct link has

also been proposed through a true polar wander–latitudinal diversity gradient (TPW-LDG) theory[18,21]. It suggests that continents shifting equatorward, i.e., moving into the LDG, would experience enhanced origination and hence diversity increase, while those shifting poleward, i.e., moving out of the LDG, would experience enhanced extinction and hence diversity decrease. This quadrature pattern of TPW effects on diversity is similar to those predicted for relative sea-level change during TPW[25], which can further amplify the anticipated diversity changes through sea-level-related artifacts, e.g., continental margins moving equatorward should experience elevated origination (though TPW-LDG theory), and additionally, fossils of these new species are more likely to be preserved due to the concomitant transgression in sea level[21]. Recently reported high-resolution biodiversity records[1,7,11] suggest a protracted two- or three-phase extinction lasting from the Katian stage to the Hirnantian stage (Fig. 6c, d), rather than a two-pulse extinction limited to the Hirnantian[5]. This modified extinction pattern may indicate that previous kill mechanisms proposed to fit the two-pulse extinction situation have become at least partially invalidated or weakened[59].

Our reconstructions demonstrate that during 450–445 Ma, TPW moved southern Gondwana poleward and northern Gondwana equatorward (Fig. 5), with more area of Gondwana on average shifting poleward (Fig. 5a, b). This observation, according to TPW-LDG theory, would cause both enhanced origination and extinction, but with extinction overwhelming origination. This palaeogeographic prediction appears to be supported by the fossil record[11] (Fig. 6d). After that, a radiation phase (Fig. 6c, d) should correspond with the continuing equatorward shift of Gondwana, Baltica, and Turan–Karakum–Tarim–North China until the early Hirnantian. Thus, during the fleeting Hirnantian stage, both the increased tropical weathering of arc-continent collisions triggered glaciation and the further poleward shifting of Siberia, Turan–Karakum–Tarim–North China, South China and south and east Gondwana, caused the second severe extinction (Fig. 6c, d). Paleogeography after the TPW event also favored plant colonization as a majority of continents became located at low-to-mid latitudes (Fig. 5c), which is supported by the positive carbon isotope signal (Fig. 6b) and oxygen rise during this time[7]. Overall, the proposed TPW and the palaeogeographic reorganization resulting from the ~50° reorientation provide a simple and basic mechanism for the dramatic environmental changes of the end Ordovician early Silurian. These connections reflect the intimate coupling between the evolution of Earth's spheres: TPW is induced by changes of subducting slabs in the mantle; in turn, TPW resulted in palaeogeographic changes that influenced Earth's hydrosphere, cryosphere, and biosphere.

## Methods
### Sampling
Our sampling sections are in Xiushan County, east Chongqing, China (Supplementary Figs. 1, 2). Silurian strata in this area were folded during the middle Mesozoic (Jurassic–Cretaceous)[67]. Ascending in stratigraphic order, the Silurian strata consist of the Llandovery Longmaxi Formation (Fm) black shale, the Xiaoheba Fm green siltstone, the Rongxi Fm red beds, the Xiushan Fm siltstone, the Huixingshao Fm red beds, and the Ludlow-Pridoli Xiaoxi Fm siltstone with some red beds (Supplementary Figs. 1, 2). There are disconformities between the Silurian strata and both its overlying Devonian and underlying Ordovician strata (Supplementary Figs. 1, 2).

As previous studies[32,35] had intensively sampled the Rongxi Fm (Supplementary Fig. 1), we conducted our palaeomagnetic study on the previously sparsely sampled Huixingshao Fm at three sections (Supplementary Figs. 1, 2). The section Yongdong (SY) is on the west limb of a steeply dipping syncline (GPS: 28.610°N, 109.157°E; Supplementary Fig. 2). Six sites about 104 samples are collected here. The sections Tianlu and Kapeng (ST and SK) are on the east limb of the same syncline (ST GPS: 28.551°N, 109.162°E; SK GPS: 28.631°N,

109.287°E; Supplementary Fig. 2). One site (10 samples) and four sites (43 samples) were collected from the ST and SK sections, respectively. All samples were collected with a portable gasoline-powered drill and oriented with a magnetic compass.

### Palemagnetism and rock magnetism
All samples were cut into at least one standard specimen (height = 2.3 cm, diameter = 2.54 cm). Natural remanent magnetization (NRM) was firstly measured. Then, all specimens were subjected to stepwise thermal demagnetization using an ASC-TD48 oven, and remanent magnetizations were measured using a 2G-755 cryogenic superconducting magnetometer housed in a magnetically shielded room. Typically, demagnetization was applied in steps of 10–100 °C, starting at 100 °C and going up to 670 °C. Backfield demagnetization curves of representative specimens, with a 2 T saturation field followed by a progressively larger filed in opposite direction, were conducted on the VSM 8600 series (Lake Shore Cryotronics, Inc.). Susceptibility temperature ($\kappa$-$T$) experiments, heated and cooled between 30–700 °C, were measured in air by using the KLY4 Kappa bridge (AGICO) devices. All experiments were done at the Key Laboratory of Paleomagnetism and Reconstruction, Ministry of Natural Resources, Beijing.

### Probability of sampling true polar wander
For any quantity, $\varphi$, that varies as a function of time, the probability density function (PDF) describes the relative amount of time spent at each value of $\varphi$ or, alternatively, the likelihood that a particular value of $\varphi$ will be sampled. For TPW, we let $\varphi(t)$ be the angular separation of the rotation axis relative to a fixed geographic axis (TPW angle), or the angular distance from one estimate of the location of the spin axis to some later estimate. An un-normalized PDF for $\varphi(t)$ is then given by

$$f(\varphi) \equiv \left| \frac{dt}{d\varphi} \right| = \frac{1}{|d\varphi/dt|}, \tag{1}$$

so that the probability of sampling φ within a given angular interval is

$$P(\varphi_1 < \varphi < \varphi_2) = \alpha \int_{\varphi_1}^{\varphi_2} f(\varphi) d\varphi, \tag{2}$$

where $\alpha$ is a constant chosen so that the total probability $P(0° < \varphi < 90°)$ equals 1. Equation (1) shows that the likelihood that φ is sampled is exactly inversely proportional to the TPW speed, $|d\varphi/dt|$. This straightforward relationship between TPW sampling probability (PDF) and TPW speed shows that faster instantaneous TPW speeds are prone to be undersampled. Technically speaking, the uphill battle to observe rapid TPW is nothing more than the age-old stroboscopic effect, or aliasing: the difficulty of studying rotating planets, reciprocating blades, oscillating fans, or vibrating strings with discrete data; temporal undersampling is a major hurdle for understanding geologic processes[68] and TPW is not an exception.

We now demonstrate an example of the undersampling problem with a simple modeled TPW excursion. In the following example, we use the simple analytical framework of Tsai and Stevenson[69] to describe the TPW due to a chosen moment of inertia tensor variation. In this formulation, the Liouville equation for a viscoelastic planet is analytically solved for a given perturbation of the moment of inertia tensor, following Munk and MacDonald[70], to obtain the TPW angle. For simplicity, we chose a moment of inertia variation that is sinusoidal with a period of 150 Ma, with an average viscosity of $3 \times 10^{22}$ Pa s, and an amplitude of $10^{-5}C$ (where $C$ is the Earth's moment of inertia). This variation is chosen to very roughly approximate the observed TPW described in the next section. Equation (12) of Tsai and Stevenson[69] then yields the TPW angle as a function of time, $\varphi(t)$, which is plotted in Fig. 4a for an arbitrary chosen initial condition. The associated TPW speed (black line) and the PDF (shaded gray) for this TPW curve are plotted as a function of time in

Fig. 4b. As shown, the maximum instantaneous TPW speed is about $2.2° \text{ Ma}^{-1}$ (24 cm yr$^{-1}$) and is associated with a minimum in the PDF, showing that it is the least probable value to be observed if one randomly samples the distribution. One can also compare the probability of sampling within a finite range by using Eq. (2), or by simply reading time intervals from Fig. 4a. For example, the values of $\varphi$ in the range $4° < \varphi < 9°$ represent ~32% of all measurements whereas a similar 5° range $40° < \varphi < 45°$ represents only about 1.5% of all measurements. For this example, then, it is about 20 times more likely to sample within the first range of $\varphi$ (low instantaneous TPW speed) compared with the second range (maximum instantaneous TPW speed).

## Data availability
The palaeomagnetic data generated in this study, including the magnetometer measurements and backfield demagnetization data, have been deposited in the Open Science Framework database (https://osf.io/z2cds/). Palaeomagnetic statistical data, palaeopoles and Euler parameters used in this study are provided in the Supplementary information.

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

## Acknowledgements

We thank Nie Lei and Chen Yang of Chongqing Key Laboratory of Exogenic Mineralization and Mine Environment, Chongqing Institute of Geology and Mineral Resources for information about the sampling sections and assistance to the drafting of Supplementary Fig. 2. Victor Tsai provided Fig. 4 depicting the probability of sampling TPW. This study was supported by the Natural Science Foundation of China (grants 91855216, 41230208, 42002208). X.J. was supported by the China Scholarship Council (no. 201808110036). R.M. was supported by the Natural Science Foundation of China (grant 41888101) and the Key Research Program of the Institute of Geology & Geophysics, CAS (grant IGGCAS-201905).

## Author contributions

X.J., R.M., Z.Y. concept this research. X.J., Z.Y., M.Z., Y.T. conducted the field investigation and sampling. X.J. implemented all the thermal demagnetization and rock magnetic experiments. X.J., R.M., Z.Y., B.W. analyzed the data. Figures 1l, 3b, 5a–c, Supplementary Fig. 2 and Supplementary Fig. 5 were generated by X.J. Snowflake in Fig. 6a was generated by X.J. Map used in Supplementary Fig. 1 was generated by X.J. R.M. drew the Fig. 4. Figures 3, 5, 6 were generated by X.J. and R.M. X.J., R.M., Z.Y. wrote the original draft. All authors contributed to the manuscript rewriting and editing.

## Competing interests

The authors declare no competing interests.
