## [Peer Review File · Nature Communications]

Ordovician–Silurian true polar wander as a mechanism for severe glaciation and mass extinctionReviewer #1 (Remarks to the Author):

Review by Richard G Gordon of "Ordovician-Silurian true polar wander as a mechanism for severe glaciation and mass extinction" Xianqing Jing et al.

This is an interesting paper based on some new data and some bold ideas. On balance I think that the heart of the paper is excellent and ought to be eventually published in a widely read journal, but there are more than a few flaws that need to be addressed and corrected first.

Some positive comments: The tying together of true polar wander together with severe glaciation, mass extinction, and species origination in the manuscript is bold and fascinating. It is hard to imagine how $\sim 60^\circ$ of apparent polar wander (APW) in ~ 10 Ma could be caused by anything but true polar wander (TPW), which supports their thesis. They do a reasonably good job of setting the context of the end-Ordovician mass extinction ("...the second most lethal of the 'Big Five'...").

Some flaws and weaknesses: (1) The authors falsely conflate rates of APW with rates of continental motion. In general, the rate of continental motion will be lower, conceivably lower by an order of magnitude or more. The authors don't really give us the context for the glaciation. (2) They should remind the reader of what percentage of Earth history is thought to have been an "ice-house Earth" and what percentage was "greenhouse of hothouse Earth" to give some context to a wider audience. Also, at the least they should say something to the effect that the "glaciation is the second of five occurrences of significant land ice or the state of an ice-house Earth during Phanerozoic time". (3) The weak link of their thesis is the observed APW of Laurentia, which does not exhibit the 60° of APW in 10 Ma exhibited by the other continents. Their argument that it is due to plate motion such that it cancels out the effect of TPW is not convincing for the reason that the indicated plate motion is inconceivably fast. While this could cancel out some of the effect of TPW, most of the nominal-difference has to be attributed to noise or uncertainty, which hopefully can be tested more rigorously by future paleomagnetic investigations of Laurentian APW. (4) There are a number of lesser problems that need to be addressed. In particular the authors in many places make assertions without presenting the evidence or logic behind the assertions. In all these places the evidence for these assertions requires a full elaboration. Details are included in the remarks below. (5) The authors incorrectly write "Early Silurian" and "Late Silurian" in many places. "early" and "late" should not be capitalized as these are not formally recognized divisions of geologic time.

Line-by-line comments:

Line 9: This is the rate of APW, NOT the rate of motion of the continents. The relation between a segment of an APW path and the minimum velocity of the continent was worked out by Gordon, McWilliams, and Cox 1979 (Journal of Geophysical Research). The rate of continental motion can be much lower than the rate of APW.

Line 10: Sentence needs to be rewritten, specifically "contradicts plate tectonics" makes no sense to me. You should think more specifically about what you think the contradiction is and write it instead. I think that what you might mean that the very large rate of implied net rotation of the lithosphere relative to the mantle is physically implausible. In any event, what you have written makes no sense.

Line 14: Readers won't know what you mean by the "global quadrature pattern of continental motion during TPW", but I guess they can carefully read the paper to find out. It would be better if you could write this in a way that the readers will understand just from the abstract. I also find the use of the term "continental motion" confusing because geologists usually use this for motion relative to a mean mantle reference frame, not the spin axis. So I think it might be better to write "pattern of latitude change during TPW" rather than "pattern of continental motion during TPW".

Line 62: Near here would be a good place to put the O-S glaciation in context in the Phanerozoic.

Line 67: Need a comma after "continents".

Line 68: re: "relative to the liquid core". I am aware that several scientists have written something like this in prior papers, but it makes little sense to me. We don't really know whether the liquid core participated in the rotation or not. We do know that the solid

earth re-oriented relative to the spin axis, however, and that's the right way to express this I believe.

Line 81: re: "period". This word should not be used in geologic writing except for "Period", which is a formally recognized division of geologic time. Use "time interval" or "interval" instead.

Line 93: Instead of the qualitative term "precisely", it would be more useful to the reader if you quantified the degree of precision. Did the events occur within 10,000 years of on another? 1,000 years? 100,000 years? 1,000,000 years? As it stands the reader has no idea what "precisely" means here.

Line 101: I think it would be useful to state here the duration of the Silurian Period to give some context to a "mean pole for the whole period".

Line 105: "data" is plural, so change "was" to "were".

Lines 128-130: Given the 10-Ma age gap, $64^{\circ} \pm 8^{\circ}$ does imply an impressive rate of APW. It does not, however, imply a similarly high rate of continental motion. This is sloppy thinking that needs to be corrected.

Lines 140-141: This repeats the logical error made above. These are rates of APW and there is no justification given for interpreting them as rates of continental motion. (The implied rates of continental motion are much much lower.)

Lines 141-142: This argument is incorrect because you have not established that the continents have moved this fast. To make this argument, you should apply an analysis like that of Gordon et al. 1979. Pre-Tertiary velocities of the continents: A lower bound from paleomagnetic data, RG Gordon, MO McWilliams, A Cox, Journal of Geophysical Research: Solid Earth 84 (B10), 5480-5486. Such an analysis will almost surely show that the minimum speeds of the continents required by the paleomagnetic data are much much lower than the ones you give in the manuscript.

Lines 145-146. The authors are conflating two different things, relative plate motion ("seafloor spreading") and absolute plate motion (typically motion relative to a mean mantle reference frame). To me, this sentence makes no sense.

Lines 146-150: I don't think this sentence adds anything to the discussion.

Lines 150-151: This assertion is not justified. For example, examine the confidence limits of the locations of "paleomagnetic Euler poles" in the work of Gordon et al. 1984 (Tectonics). In the two examples they consider, one is consistent with a great circle and the other is not. Both examples are consistent with an infinite number of small circles of varying radii, which I assert is true of all APW tracks. So the assertion is flat-out wrong and the sentence needs to be rewritten.

Lines 151-152: This assertion is wrong because its premises are wrong.

Line 154: "plate tectonics" is vague and I can only guess what the authors mean. Maybe they mean something like "the motions of the plates on which these five continents are passive passengers is unlikely to be capable of explaining the observed APW without invoking TPW".

Lines 156-159: I don't buy this assertion. The authors need to back it up. How much hotter do they think the mantle was in Silurian time (and give some physical basis for that answer). If one plugs that temperature difference into laboratory determined flow laws appropriate for the upper mantle, how much would that change its viscosity?

Line 167: Paleomagnetists typically use the word "stillstand" rather than "standstill" in this context.

Line 168: "compared with" rather than "compared to" is the correct preposition.

Line 187: "in fact" is superfluous. Delete it.

Line 188: What are the uncertainties on this arclength?

Line 196: Again the authors write "...while inconsistent with tectonics...". Again, there is no substance to this wording. What do the authors really want the readers to understand here?

Lines 199-200: "...is therefore predicted to circumscribe a great-circle APW path...". This is asserted without evidence or any backing by a dynamical argument and therefore can and should be dismissed by the reader. I can think of a set of simplifying assumptions that might lead to this prediction (e.g., (i) plate motion relative to the mantle is negligible, the change in the orientation of the principal axes of non-hydrostatic moment of inertia is instantaneous, (iii) those subsequently do not change at all), but that's the author's job, not the reader's. This erroneous assertion is repeated several times further

below in the manuscript.

Lines 202-203: "These systematic gaps in the APW paths of all continents are consistent with the stroboscopic effect expected for TPW....". This is another assertion made, but backed by no evidence. If the authors believe this is true, they should quantify it. What are the odds that they've captured directions 60° apart separated by 10 Ma, but not captured any pole positions during that 10 Ma on 5 different continents? My guess is that the systematic gaps are not consistent at all. It's up to the authors to present evidence that they are consistent and to quantify the argument probabilistically.

Line 212: Any complete discussion of mass anomalies should include the waxing and waning of ice sheets.

Line 265: It's my understanding that relative paleo-longitude can be determined this way, but not absolute paleo-longitude. This should be clarified in the manuscript.

Reviewer #2 (Remarks to the Author):

Reviewer #2 Attachment on the following page

Reviewer-report of the paper “Ordovician–Silurian true polar wander as a mechanism for severe glaciation and mass extinction” by Jing et al., manuscript submitted to Nature communications.

The paper by Jing et al. (“Ordovician–Silurian true polar wander as a mechanism for severe glaciation and mass extinction”), submitted to Nature communications, is well-written, presenting new paleomagnetic data and a (not totally new) idea of enhanced plate velocities during the 450 – 440 Ma interval, what the authors interpret as a signal of True Polar Wander (TPW). This hypothesis is undoubtedly of wide interest for the Paleozoic paleogeography and Earth geodynamics. While the TPW analysis is well-crafted and solid, my review will first focus on the quality of the new paleomagnetic results (the Huixingshao pole, and the recalculated Rongxi pole) presented in this manuscript. Indeed, to investigate rapid true polar wander it is necessary to obtain well-constrained paleomagnetic poles considered robust by the community. Such high-quality pole should have a prerequisite to meet the Nature Communications criteria for publication. Unfortunately, these new paleomagnetic data from the Huixingshao Fm. suffer from a lack of evidence to confirm the primary character of the remanence (no positive fold test). The absence of such a prerequisite is therefore the major problem I identify, which therefore makes it problematic to use these data for the main topic of the paper: the TPW analysis. I think the authors can overcome this concern after major modifications to improve these results. Here, I will use the line numbers, pointing to the questions and problems. I will give some advice to improve the Figures and Tables.

Results

The authors provide the results of acquisition of novel paleomagnetic data from three sections of the Huixingshao Formation (1) in Xiushan county, Chongqing, South China. In addition, they recalculated the Rongxi pole (2) using the data of Opdyke et al. (1987) and Huang et al. (2000). To assess the quality of these new results, I will use the refined R-criteria of Meert et al. (2020), which now supersedes the Van der Voo (1990) factor (Q) used by the authors:

R1 ~Well-determined rock age and a presumption that magnetization is the same age

- (1) The Huixingshao Fm. is estimated at ~436–435 Ma (Line 104).
- (2) The older Rongxi Fm. is estimated at ~438–437 Ma (Line 120).

As no radiochronometric data are available, the ages of these units are only estimated using a poorly biostratigraphical control and some uncertainties exist whether the age of the Rongxi Fm. is Telychian or Aeronian (Rong et al., 2019). Based on available data, the Telychian age used in this paper seems reasonable (Huang et al., 2017) and will need to be confirmed in the future.

R2 ~Techniques and Statistical analysis

- (1) Based on only 6 sites, the Huixingshao pole does not meet this criterion and the minimum of 8 sites required to calculate a paleomagnetic pole. In addition, the precision parameter (K) for the VGP distribution is ~90.8 (not shown in the Table 1 supp. data), which could indicate some suspicion of inadequate averaging of secular variation (>70; (Meert et al., 2020)). As this issue was not discussed in the paper, I strongly suggest to the authors to provide the Deenen et al. (2011) confidence bounds.

(2) The recalculated ~438 Ma Rongxi pole fulfills the R2-criterion but no details were performed by the authors about the used sites of Opdyke et al. (1987) and Huang et al. (2000):

Line 123-125: the recalculated Rongxi pole was calculated using the the Tongzi, Dagan, and Shiqian sections of Opdyke et al. (1987) and Huang et al. (2000). This was repeated in Line 38-39 of the supplementary data and the name **SDT** was used in the Figures for this pole (**Shiqian, Dagan, Tongzi**).

In the Supplementary Fig. 5b, the site-mean direction for the Rongxi Fm. is illustrated, and particularly the C1 component of Huang et al. (2000), considered as primary. The results and the Fisher 's statistics are indicated in orange. I was able to check and I get exactly the same values and the same stereo using:

- the sites 8,9,10,11,12,13,14 from Shiqian (Huang et al., 2000)
- the site a from Tongzi (Opdyke et al., 1987)
- the sites J, K,L,M,N,O,P,Q,R,S,T,V from the Xiushan section (Opdyke et al., 1987)

The associated paleomagnetic pole with this mean direction is located at 10.3°N, 196°E.

This Rongxi pole is different from the pole used in Table 2 (-17.9°N, 20.3°E). Although I was not able to obtain exactly these values, I obtained a close result by using the sites of Shiqian, Dagan, and one site of Tongzi (*i.e.*, the SDT acronym) ...

Therefore, I don't understand why to use in the supplementary Fig. 5b the Xushian sites and not the Dagan sites and finally in the recalculated Rongxi pole of Table 2 the Dagan sites. In addition, the Dagan sites were considered as C2 component by Huang et al. (2000) and interpreted as a "remagnetization" component (see the caption of the Fig.5b in the supplementary data). If you have decided to use the Dagan sites as primary in the calculation of your new Rongxi pole (SDT), then you must absolutely justify this choice.

In short, the calculation of your new ~438 Ma Rongxi paleopole is terribly confusing and only a table can help to clarify it!

R3 ~Evaluation of remanence carriers

In addition to the supplementary Fig.3, stereographic projections and magnetization intensity decay curves for some examples will be useful to see the behavior of these samples. It is important to show these curves because you mention the high unblocking temperature at the Line 108 (main text) and also in the Lines 53-54 (supplementary data) to justify the hematite

as the main remanent carrier. Fig.8 in the supplementary data is welcome. Since the authors suggest that the SK and ST sections are remagnetized, it would be important to show whether there are differences in magnetic mineralogy between these sections.

R4 ~Field Tests that constrain age of magnetization

- (1) Unfortunately, no fold test is available for the Huixingshao pole although 3 different sections were sampled from each side of the syncline. Beyond the small number of sites used, this does not allow to consider this pole as robust and therefore the question arises if it can be used to characterize the position of the South China Block.
- (2) As outlined above, the calculation of the newly Rongxi pole must be reviewed, and it is important to point out if the chosen sites pass a fold test.

R5 ~ Structural control, and tectonic coherence with craton or block involved. Inclination shallowing assessed in clastic sedimentary rocks

Using paleomagnetic comparison between the Rongxi data from the Xiushan section (Opdyke et al.,1987) and the Rongxi data from the Shiqian/Daguan/Tongzi (SDT pole), the authors suggest a $12.1 \pm 3.9^\circ$ clockwise rotation (Fig.7. supplementary data). First, I cannot check this value without to be sure of the used sites for the SDT pole as previously discussed.

One question I have is why use the Daguan sites (C2 component) in the calculation? It's interesting to see that these are the sites that pull the two directions apart. Moreover, the Daguan sites are geographically very distant which cannot exclude a complex tectonic history between these areas? Huang et al. (2000) have seen that the directions (C1) of Shiqian were similar to those of Xushian and this was an important argument that allowed him to suggest that the C1 component was indeed primary! According to this statement there was no need to put a rotation of 12° between the sites of Xiushan and Shiqian for the Rongxi Fm as it is proposed in this paper.

In addition, the authors use this rotation to correct Huixingshao Fm pole of the Yongdong section (SY). This would correspond to a different history and a very local rotation (evidence of that?). Perhaps the fault marked in Fig. 1 but all of this remains highly speculative.

Another comment about this rotation is that the authors claim that the SK and ST section are remagnetized, contrary to the SY section. But by looking in detail, the SK-ST direction share a common direction with the SDT direction (McFadden and McElhinny, 1990). It is not clear why these sites (SK-ST) are excluded contrary to the Yongdong (SY) section.

Therefore, is the rotation proposed by the authors not an artifact related to the choice of pmag sites used and not a tectonic problem? I ask this question and I think that additional data should be provided (structural data, AMS, fold test to prove which limb are remagnetized...). In view of the elements, neither of the two poles can validate this R5 criterion.

R6 ~The presence of magnetic reversals

Lines 111-113: As illustrated in Fig.5a (supplementary data), two antipodal directions were revealed for the Huixingshao Fm with a positive reversal test (C-class) according to McFadden and McElhinny (1990). This argument is the only one left to justify the primary character of the high temperature component since there is no fold test. It should be remembered that a remagnetization with two-polarities is also possible. Interesting, the authors provide a magnetostratigraphy of the Yongdong section for the Huixingshao Fm. (Fig. 4 supplementary data). Clearly, we can see the reversal state of the SY8 site. The second reversed site is the SY10 as illustrated in the Fig.5a and listed in the Table 1 (supplementary data). For the SY10 site, we cannot clearly see a reversion in the declination and inclination curves! Besides, the

authors used for this site the expression “normal with reversal samples”. In the inclination curve I can see only one specimen in the reversed polarity. This seems to be in total contradiction with the Table 1 where I can read 20 specimens showing a site-mean inclination of -39.6°... I therefore ask the question if there was not only one specimen in reverse polarity and that the authors have changed the polarity of all other specimens in normal polarity. If this is the case, then there would be only one zone of true reverse polarity in the section SY and the reversal test would be inconclusive. We need to be sure about that and the magnetostratigraphy needs to be coherent with the site-mean directions.

R7 ~No resemblance to younger poles by more than a period based on overlapping A95

Fig.6 supplementary data show that the SDT pole and the Huixingshao pole are different from younger poles to fulfill this R7-criterion.

In summary, we can assign the R-criteria in the following Table for the Huixingshao and Rongxi poles. This differs from the 7 and 6 rating using the Van der Voo’s criteria (see the Table 2). This highlights that objectively, the recalculation of the Rongxi pole is clearly more robust than the Huixingshao pole from the Yongdong section (SY) if the Rongxi passes a fold test. In order for the proposed TPW hypothesis and analysis to be convincing it is therefore necessary to insist on the use of the most robust poles, which is not the case with the Huixingshao pole.

Name	Age (Ma)	R1 (age) ~15 Ma + mag mem age	R2 (stats) mult demag method + >8 sites (3spec min/site) PSV N>25 spec, 10<K<70	R3 (rock mag)	R4 (tests)	R5 (structural control) si incl correction 0	R6 (reversals) A/B/C rated	R7 (no resemblance younger)	R	References
SOUTH CHINA										
Huixingshao Fm.	435	1	0	1	0	0	1	1	4	This study
Rongxi Fm.	438	1	1	1	1	0	0	1	5	This study

Additional comments:

Main text

Lines 65-66: *“Ediacaran–Cambrian periods, and this interval has been proposed to have experienced large-scale (60–90°) true polar wander^{26–32}.”*

For a complete bibliography of this topic, small TPW oscillations were also recently proposed by Antonio et al. (2021) during the Cambrian times (<535 Ma) that follow these large TPWs. This indeed underlines that this is a turbulent period from the paleomagnetic point of view and a further argument that this period may extend into the Silurian as proposed in this paper. The reference “27. Kirschvink, J. L. & Raub, T. D. A methane fuse for the Cambrian explosion...” will be more adapted in the discussion about the link TPW-biodiversity (Lines 355-393).

Lines 67-68: *“the movement of the entire solid Earth (mantle and crust) relative to the liquid core in order to stabilize Earth rotation”*

As the liquid core is turbulent (with convection movements > ~km/an), it should be better to use the reference for the movement of the Solid Earth the spin axis (fixed into the celestial reference frame).

Line 78: "However, no follow-up research has reexamined this TPW event".

When we mention a plausible Ordovician-Devonian TPW, I suggest adding the reference of Piper (2006) who also published on this topic 10 years later than Van der Voo (1994). Therefore, your line 78 is inappropriate.

Line 165-193: "palaeomagnetic poles from Laurentia during this time are characterized, in contrast, by much less APW, and almost essentially a standstill"...

This discussion about the motion of the Laurentia is convincing! Intriguing, Australian poles show a similar behavior during the Cambrian (<540 Ma) (Antonio et al., 2021). As Australia is in opposite position along the great circle (Fig. 2), this could suggest a fixed Imin axis position in the Early Paleozoic times..

Lines 244-....: About the paleogeographic reconstructions based on TPW.

The available paleomagnetic data for this interval exhibit abrupt directional variations, and previous efforts have been made to explain this erratic behaviour (appropriately recognized and cited in the paper). Their findings show that global APW paths imply high velocities incompatible with plate tectonics, so they appeal directly the hypothesis of large TPW events to account for the misfit. It is worth noting that this paper is missing a few sentences about the other possibilities invoked in the literature to explain these movements (collision, non-uniform Earth's magnetic field) and why the authors feel that TPW is the most appropriate solution?

Fig. 1c represents the Gondwana paleolatitude and allows to delimit the tightly timing of the TPW 450-440 Ma. To confirm such global timing, it should be added on the same Figure the curves of the 6 other continents.

This calculation was based on the Wu et al. (2020)'s compilation for Baltica/Laurentia and Gondwana. It is important to note that the Wu's approach to calculate the APW paths (weighted running mean poles) is different from a simple comparison of paleopoles as it is the case in this paper for the South China block. Such differences could provide different results in the APW paths, and it is important to compare the different methods. For example, the APW path for the Gondwana of Torsvik et al. (2012) was calculate by running mean path and a 20 Ma sliding windows. Using these models, the timing of the TPW will surely be different. A comparison of the different methods could be provided in the supplementary data.

Line 321: "reconstruction of the latest Katian, Niger"

Double space between of and the.

Line 431-432: The sampling part is not detailed enough and must be developed in the supplementary data.

- So that readers not used to the local geology can follow the text, it is necessary to improve some figures :
 - o The supplementary Fig.1 needs to be reviewed. In this Figure, we have to see the locations of the different sections cited in the text (Shiqian, Daguan

...). A simplified regional tectonic setting between these sections is important because the authors used these sections for paleomagnetic corrections.

- The acronyms used need also need to be judiciously chosen explicated in the caption to help the reader: for example the Yongdong village is SY.. it will be better to use the Section Yongdong to easily understand the acronyms.

Line 457: Be careful, the manufacturer of the KLY4 is AGICO and not ASC Scientific (it is a reseller).

Lines 463-464: I think that Paleomac was also used for the Figures and calculations (see the Fig. 5), not only the reversal test.

Supplementary data

As I just said the geological context figure must be improved. The location of the different sections will allow to easily locate the other sections mentioned in Fig.2.

Fig.3: Add stereographic projections and magnetization intensity decay curves for some examples to reinforce your paleomagnetic dataset.

Fig.4: Explain the misfit of the SY10 site (only one reversed specimen while the site is considered as reverse polarity in Table 1). In addition be careful to the typo errors (declination/inclination/meters are badly written).

Fig.5: Add a stereo for the calculation of your newly Rongxi paleopole that you will use in the paper and justify the used sites (e.g., from Daguang). It is missing the number of sites used in the calculations (N) in your Figure.

Fig.6: We need a reference in the caption for the poles of reference in red.

Fig.7: this figure needs to be coherent with the calculation of the Fig.5. Use different colors for different locations.

Fig.8 : ok!

Table 1: add columns for P.lat and P.lon rather than lines as you did. This will save space and the table will be clearer. Add the sites in the table (or a new) for your Rongxi direction, the associated mean directions and the mean pole. Add also the Huixingshao pole before and after rotation!

Table 2: Check the reference for the Tarim craton (Huang et al.1). Use the R-criteria (Meert et al., 2020) instead of the Q of Van der Voo. Add also if they are paleomagnetic poles or synthetic poles from averages (+ the method of calculation).

REFERENCES

- Antonio, P.Y.J., Trindade, R.I.F., Giacomini, B., Brandt, D., Tohver, E., 2021. New high-quality paleomagnetic data from the Borborema Province (NE Brazil): Refinement of the APW path of Gondwana in the Early Cambrian. *Precambrian Res.* 360, 106243.
- Deenen, M.H.L., Langereis, C.G., van Hinsbergen, D.J.J., Biggin, A.J., 2011. Geomagnetic secular variation and the statistics of palaeomagnetic directions. *Geophys. J. Int.* 186, 509-520.

- Huang, B., Wang, Y., Tang, P., Wei, X., Zhang, X., Wang, G., Wu, R., Zhang, Y., Zhan, R., Rong, J., 2017. On the uppermost Xiushan Formation (Llandovery, Silurian) at Jianshi area, Hubei Province. *J Strat*, 41: 375-385.
- Huang, K., Opdyke, N.D., Zhu, R., 2000. Further paleomagnetic results from the Silurian of the Yangtze Block and their implications. *Earth Planet. Sci. Lett.* 175, 191-202.
- McFadden, P.L., McElhinny, M.W., 1990. Classification of the reversal test in palaeomagnetism. *Geophys. J. Int.* 103, 725-729.
- Meert, J.G., Pivarunas, A.F., Evans, D.A.D., Pisarevsky, S.A., Pesonen, L.J., Li, Z.-X., Elming, S.-Å., Miller, S.R., Zhang, S., Salminen, J.M., 2020. The magnificent seven: A proposal for modest revision of the Van der Voo (1990) quality index. *Tectonophysics* 790, 228549.
- Opdyke, N.D., Huang, K., Xu, G., Zhang, W., Kent, D.V., 1987. Paleomagnetic results from the Silurian of the Yangtze paraplatform. *Tectonophysics* 139, 123-132.
- Piper, J.D.A., 2006. A ~90° Late Silurian–Early Devonian apparent polar wander loop: The latest inertial interchange of planet earth? *Earth Planet. Sci. Lett.* 250, 345-357.
- Rong, J., Wang, Y., Zhan, R., Fan, J., Huang, B., Tang, P., Li, Y., Zhang, X., Wu, R., Wang, G., Wei, X., 2019. Silurian integrative stratigraphy and timescale of China. *Science China Earth Sciences* 62, 89-111.
- Torsvik, T.H., Van der Voo, R., Preeden, U., Mac Niocaill, C., Steinberger, B., Doubrovine, P.V., van Hinsbergen, D.J.J., Domeier, M., Gaina, C., Tohver, E., Meert, J.G., McCausland, P.J.A., Cocks, L.R.M., 2012. Phanerozoic polar wander, palaeogeography and dynamics. *Earth Sci. Rev.* 114, 325-368.
- Van der Voo, R., 1990. The reliability of paleomagnetic data. *Tectonophysics* 184, 1-9.
- Van der Voo, R., 1994. True polar wander during the middle Paleozoic? *Earth Planet. Sci. Lett.* 122, 239-243.
- Wu, L., Murphy, J.B., Quesada, C., Li, Z.-X., Waldron, J.W., Williams, S., Pisarevsky, S., Collins, W.J., 2020. The amalgamation of Pangea: Paleomagnetic and geological observations revisited. *GSA Bulletin*.

Reviewer #3 (Remarks to the Author):

Review of NCOMMS-22-16939 'Ordovician-Silurian true polar wander as a mechanism for severe glaciation and mass extinction'

This paper presents new paleomagnetic data from South China that dates from 440-450 Myr ago. Combined with paleomagnetic data from the other major continents that existed at this time, this paper argues for a rapid true polar wander event, and uses this as a basis to explain glaciation and mass extinction at the Ordovician-Silurian transition.

The paper appears to present a new dataset and comprehensive analysis that illuminates a long-standing issue in a particularly satisfying way. Moreover, the paper is well written and logically organised, making it easy to follow. All of the arguments appear sound and the data support the conclusions and claims.

My background is in paleointensity determination from challenging samples, and I have very little experience in plate reconstructions and the impact of plate motion on the surface of the Earth. With this in mind, I am not able to fairly review either the plate reconstruction or the mass extinction/glaciation aspects of this paper. I hope that the other reviewers have been chosen to focus more on these aspects of the paper, or all the aspects of the paper in combination. Instead, I am able to focus on the paleomagnetic measurements themselves, and my review will focus almost entirely on these.

Overall, the paleomagnetic data are collected using a tried-and-tested technique, and everything appears to have been conducted in a reliable, standard manner. As such, I have no comments on the experimental approach. My main comments relate to the presentation of the paleomagnetic data:

-There is no actual data in the main text of the paper, making the raw data difficult to examine or scrutinise, which are the basis of all the later arguments. If permitted by the number of figures allowed, please add at least one of figures S3, S4, or S5 into the main text.

-None of the axes in Fig S3 have values on them, and only the vertical axis has a label. Please add values and labels.

-The orthogonal project plots in Fig 3S are said to be 'representative' in the caption, but there are no data from SY11, SY12, or SY13 — this means there are no raw data from these sites in the paper at all. Moreover, SY10-1B and SY10-14B are included in the orthogonal projection plots despite not appearing to be representative of SY10 as a whole. Further discussion on these two datasets would be appreciated. For instance, they have very different inclinations, so it would be good to see more clarity on which data went into calculating the average datapoint for the SY10 point in Fig 4a that is similarly orientated to SY8. With this in mind, is it fair to call them reversed samples, given only one of them has an orientation close to the points from SY8.

-A lot of the data presented is limited to final, average directions. While this is the key information that enables the plate reconstructions to be conducted, it would be more transparent to include all of the data in Fig S4 as well as the averages. Personally, I like to see this in a paper, as it enables readers to assess the data for themselves.

-Finally, the low-temperature component is said to be a VRM consistent with the direction of the present-day local geomagnetic field. While the LT component in several of the ST and SK samples are to have similar orientations, this doesn't appear to be the case for SY10-1B and SY10-14B, or SK2-2. Perhaps an equivalent of Fig S5 for the LT component with the present-day local field orientation would help to demonstrate how many samples do carry an LT component with the same direction as the present day field.

This would hopefully help build confidence in the data.

1. Responses to Reviewer # 1

Q1: They should remind the reader of what percentage of Earth history is thought to have been an “ice-house Earth” and what percentage was “greenhouse of hothouse Earth” to give some context to a wider audience. Also, at the least they should say something to the effect that the “glaciation is the second of five occurrences of significant land ice or the state of an ice-house Earth during Phanerozoic time”.

R: We appreciate the reviewer’s suggestion to provide additional context for the O–S glaciation. In response to the reviewer’s suggested additional text above, we would note that an earlier Phanerozoic glaciation during the Cambrian has been hypothesized from one locality (the small microcontinent of Avalon(Landing and MacGabhann, 2010), and has yet to be corroborated by other workers or identified on other larger continents. Thus, even if viable locally, the middle Cambrian glaciation, as of yet, does not meet the reviewer’s criteria of “significant land ice or the state of an ice-house Earth during Phanerozoic time.” We have thus added additional text provided more context for the O–S glaciation, but still assert that it is the first major glacial interval of the Phanerozoic (lines 32-37):

“Accompanying the O–S mass extinction was the first of five occurrences of significant glaciation during the Phanerozoic Eon (ca. 541 Ma to present), of which only ~25% represented glacial intervals. The O–S glacial interval is unusual both because it was short-lived and occurred at a higher atmospheric partial pressure of CO₂, perhaps 8–16 times higher than today.”

Q2: The weak link of their thesis is the observed APW of Laurentia, which does not exhibit the 60° of APW in 10 Ma exhibited by the other continents. Their argument that it is due to plate motion such that it cancels out the effect of TPW is not convincing for the reason that the indicated plate motion is inconceivably fast. While this could cancel out some of the effect of TPW, most of the nominal-difference has to be attributed to noise or uncertainty, which hopefully can be tested more rigorously by future paleomagnetic investigations of Laurentian APW.

R: We strongly agree with the reviewer that future paleomagnetic investigations of Laurentian APW are important to test our interpretation of superimposed rapid TPW and rapid plate motion during the closure of the Iapetus Ocean. Slightly contrary to the reviewer’s representation of our interpretation, we do not argue that the *entire* ~50° TPW event is masked by superimposed plate motion of Laurentia. As detailed in the main text (because we agree with the reviewer of addressing its importance as a caveat of our study), we conducted a statistical *F*-test that demonstrates that the 18.9° difference between Laurentia poles, despite their large pole uncertainties, is nonetheless statistically significant. Therefore, Laurentia poles do exhibit ~20° of APW and thus only ~30° of the TPW event are potentially offset by simultaneous plate motion. This would imply more realistic plate motion speeds for Laurentia (~3° Myr⁻¹, or ~33 cm yr⁻¹) than the reviewer suggested, and we also note that motion of Laurentia at ca. 1.1 Ga as fast as ~30 cm yr⁻¹ can potentially be interpreted as plate motion(Swanson-Hysell et al., 2019). Nonetheless, we personally agree with the reviewer that

even this rate appears rather fast for plate tectonics. Thus, in the revision we make it clear that data from Laurentia are not currently accurate enough (few poles with large errors) to confidently calculate how much TPW may have been canceled out exactly. We just propose an explanation for these data available to us now. We add this mention of the data-error explanation in lines 228-229:

“Otherwise, this relative stillstand may be an artifact of the large age errors of these poles used for APW comparison”

Q3: The authors incorrectly write “Early Silurian” and “Late Silurian” in many places. “early” and “late” should not be capitalized as these are not formally recognized divisions of geologic time

R: all have been changed.

Q4: Line 9: This is the rate of APW, NOT the rate of motion of the continents. The relation between a segment of an APW path and the minimum velocity of the continent was worked out by Gordon, McWilliams, and Cox 1979 (Journal of Geophysical Research). The rate of continental motion can be much lower than the rate of APW.

R: We thank the reviewer for pointing out our misrepresentation about APW vs. continental rates. Indeed, depending on where a continent is positioned relative to the Euler pole implied by the APW, the rate of continental motion will differ from the APW rate. That is, only when a continent is $\sim 90^\circ$ away from the Euler pole does the continental rate of motion equal APW rate; thus, in most cases, continental motion rate will be slower than APW rate. Nonetheless, our manuscript does involve the motion of West Gondwana that is in the plane of APW and thus $\sim 90^\circ$ away from the Euler pole of motion. Thus, it is fair to say that “maximum” speeds of continental motion can reach the observed APW rate. Also, in most instances we are interested in comparing APW paths of different continents and thus we are generally discussing APW rate, not continental rate. We have therefore modified most instances of discussions of rates to clarify that we are discussing APW rates. In those few instances where we mention continental motion rates, we are careful not to conflate APW rates with continental motion rates in accordance with the reviewer’s accurate guidance. Below we provide two examples, one where we have revised the text to make clear we are discussing APW rates and one where we discuss continental motion but are careful to clarify we are cognizant of the difference between APW rate and continental motion rate.

Lines 169-171:

“the late Ordovician–early Silurian palaeopoles from Tarim, Siberia, Baltica, and Gondwana also all demonstrate large arc distances of APW: $54^\circ \pm 9^\circ$, $47^\circ \pm 17^\circ$, $55^\circ \pm 14^\circ$, and $58^\circ \pm 21^\circ$, respectively (Fig. 3a, Supplementary Data 2), with associated APW rates of $5.4^\circ \pm 0.9^\circ$, $4.7^\circ \pm 1.7^\circ$, $5.5^\circ \pm 1.4^\circ$, and $5.8^\circ \pm 2.1^\circ \text{ Myr}^{-1}$, respectively.”

Lines 8-10:

“Collectively, a $\sim 50^\circ$ wholesale rotation with maximum continental speeds of $\sim 55 \text{ cm yr}^{-1}$ is demonstrated.”

Q5: Line 10: Sentence needs to be rewritten, specifically “contradicts plate tectonics” makes no sense to me. You should think more specifically about what you think the contradiction is and write it instead. I think that what you might mean that the very large rate of implied net rotation of the lithosphere relative to the mantle is physically implausible. In any event, what you have written makes no sense.

R: We appreciate the reviewer for pointing out that our refutation of plate tectonic interpretation in the Abstract had not been specific enough. We have thus revised the sentence in question—within the space limitations—to provide, in the first clause, specific observations that are inconsistent with plate tectonics (e.g., unidirectional motion of separated continents, whereas plate tectonics would typically predict *relative* motion for separated continents; also plate tectonic motions need not be synchronous) or implausible for plate tectonics (e.g., the maximum rates of continental motion greatly exceed plausible plate speeds). The sentence in question now reads (lines: 10-12):

“Multiple isolated continents moving rapidly, synchronously, and unidirectionally is less consistent with and plausible for relative plate motions than TPW.”

The rest of the abstract was edited to honor space restrictions in light of the extra detail provided in the revision above.

Q6: Line 14: Readers won’t know what you mean by the “global quadrature pattern of continental motion during TPW”, but I guess they can carefully read the paper to find out. It would be better if you could write this in a way that the readers will understand just from the abstract. I also find the use of the term “continental motion” confusing because geologists usually use this for motion relative to a mean mantle reference frame, not the spin axis. So I think it might be better to write “pattern of latitude change during TPW” rather than “pattern of continental motion during TPW”.

R: We rewrite it as “pattern of latitude change during TPW”

Q7: Line 62: Near here would be a good place to put the O-S glaciation in context in the Phanerozoic.

R: To contextualize the O–S glaciation, we added the following text (lines 32-37:

“Accompanying the O–S mass extinction was the first of five occurrences of significant glaciation during the Phanerozoic Eon (ca. 541 Ma to present), of which only $\sim 25\%$ was glacial intervals. The O–S glacial interval is unusual both because it was short-lived and occurred at a higher atmospheric partial pressure of CO_2 (perhaps 8–16 times higher than today).”

Q8: Line 67: Need a comma after “continents”.

R: Sentence modified in response to another comment.

Q9: Line 68: re: “relative to the liquid core”. I am aware that several scientists have written something like this in prior papers, but it makes little sense to me. We don’t really know whether the liquid core participated in the rotation or not. We do know that the solid earth re-oriented relative to the spin axis, however, and that’s the right way to express this I believe.

R: The core-mantle boundary (CMB) is by far the largest viscosity discontinuity in the planet, plus the largest long term mass anomalies arise within the mantle, so it is for both these reasons that TPW rotation almost certainly accommodated by slip at the CMB, thus rotating the solid Earth (mantle and crust) about the liquid outer core. Nonetheless, we appreciate the reviewer’s distinction here between observables and theory and agree that defining the motion relative to the spin axis is sufficient. We have thus rewritten this part accordingly (lines 69-77):

“True polar wander (TPW) is the movement of the entire solid Earth (mantle and crust) relative to Earth’s spin axis in order to stabilize Earth rotation. It is different from the plate motion of plate tectonic theory, which proposes that tectonic plates, including continents or not, move over the asthenosphere relative to the underlying convecting mantle. Tectonic plates move in different directions and with different velocities (even in the case symmetric seafloor spreading, the directions of motion are opposite of each other). In contrast, TPW can induce wholesale polar motion of the plates unidirectionally and synchronously, thus changing palaeogeography rapidly and globally”

Q10: Line 81: re: “period”. This word should not be used in geologic writing except for “Period”, which is a formally recognized division of geologic time. Use “time interval” or “interval” instead.

R: All uses of the word “period” have been changed to “time interval” or “interval”, except when explicitly referring to the division of geological time like the Silurian Period.

Q11: Line 93: Instead of the qualitative term “precisely”, it would be more useful to the reader if you quantified the degree of precision. Did the events occur within 10,000 years of on another? 1,000 years? 100,000 years? 1,000,000 years? As it stands the reader has no idea what “precisely” means here.

R: We deleted the word “precisely” as the events being synchronous is sufficient.

Q12: Line 101: I think it would be useful to state here the duration of the Silurian Period to give some context to a “mean pole for the whole period”.

R: We added the duration of the Silurian Period in the text in line 111.

Q13: Line 105: “data” is plural, so change “was” to “were”.

R: Done.

Q14: Lines 128-130: Given the 10-Ma age gap, $64^{\circ} \pm 8^{\circ}$ does imply an impressive rate of APW. It does not, however, imply a similarly high rate of continental motion. This is sloppy thinking that needs to be corrected.

R: We realize this mistake, and have changed it to “APW rate” (lines 146-147):

“Given only ~10 million years (Myr) between these two ages, the $54.4^{\circ} \pm 6.4^{\circ}$ arc distance between these two poles indicates a rapid APW rate of $5.4 \pm 0.6 \text{ Myr}^{-1}$ for South China”

Q15: Lines 140-141: This repeats the logical error made above. These are rates of APW and there is no justification given for interpreting them as rates of continental motion. (The implied rates of continental motion are much much lower.)

R: We realize this mistake, and change them to “APW rates” in lines: 172-173 as below:

“with associated APW rates of $5.4 \pm 0.9^{\circ}$, $4.7 \pm 1.7^{\circ}$, $5.5 \pm 1.4^{\circ}$, and $5.8 \pm 2.1^{\circ} \text{ Myr}^{-1}$, respectively”

Q16: Lines 141-142: This argument is incorrect because you have not established that the continents have moved this fast. To make this argument, you should apply an analysis like that of Gordon et al. 1979. Pre-Tertiary velocities of the continents: A lower bound from paleomagnetic data, RG Gordon, MO McWilliams, A Cox, Journal of Geophysical Research: Solid Earth 84 (B10), 5480-5486. Such an analysis will almost surely show that the minimum speeds of the continents required by the paleomagnetic data are much much lower than the ones you give in the manuscript.

R: This previous description has been deleted.

Q17: Lines 145-146. The authors are conflating two different things, relative plate motion (“seafloor spreading”) and absolute plate motion (typically motion relative to a mean mantle reference frame). To me, this sentence makes no sense.

R: This previous description has been deleted.

Q18: Lines 146-150: I don’t think this sentence adds anything to the discussion.

R: This previous description has been deleted.

Q19: Lines 150-151: This assertion is not justified. For example, examine the confidence limits of the locations of “paleomagnetic Euler poles” in the work of Gordon et al. 1984 (Tectonics). In the two examples they consider, one is consistent with a great circle and the other is not. Both examples are consistent with an infinite number of small circles of varying radii, which I assert is true of all APW tracks. So the assertion is flat-out wrong and the sentence needs to be rewritten.

Lines 151-152: This assertion is wrong because its premises are wrong

R: This previous description has been deleted. This part related to Q15-18 has been rewritten (lines 183-187):

“Plate motion, driven by slab subduction and mantle convection, also cannot explain these synchronous and similar large amplitude movements of multiple isolated continents either, as it requires relative motion between different plates with different velocities (speeds and/or directions).”

Q20: Line 154: “plate tectonics” is vague and I can only guess what the authors mean. Maybe they mean something like “the motions of the plates on which these five continents are passive passengers is unlikely to be capable of explaining the observed APW without invoking TPW”.

R: This sentence has been deleted.

Q21: Lines 156-159: I don’t buy this assertion. The authors need to back it up. How much hotter do they think the mantle was in Silurian time (and give some physical basis for that answer). If one plugs that temperature difference into laboratory determined flow laws appropriate for the upper mantle, how much would that change its viscosity?

R: We added two sentences and several references to back up this assertion (lines 193-196):

“Numerical simulations suggest that a 40–50° amplitude TPW event can occur in ~10 Myr if the viscosity of lower mantle is 10^{22} Pa s. Presently lower mantle viscosity is about 3×10^{22} Pa s, while it may be 3 times lower at 450 Ma.”

Q22: Line 167: Paleomagnetists typically use the word “stillstand” rather than “standstill” in this context.

R: Done. (But we note that standstill is actually already a word used by the rest of the world.)

Q23: Line 168: “compared with” rather than “compared to” is the correct preposition.

R: Done.

Q24: Line 187: “in fact” is superfluous. Delete it.

R: Done.

Q25: Line 188: What are the uncertainties on this arclength?

R: We added the uncertainty ($\pm 19.3^\circ$) on this arclength.

Q26: Line 196: Again the authors write “...while inconsistent with tectonics...”. Again, there is no substance to this wording. What do the authors really want the readers to understand here?

R: This previous description has been deleted.

Q27: Lines 199-200: "...is therefore predicted to circumscribe a great-circle APW path...". This is asserted without evidence or any backing by a dynamical argument and therefore can and should be dismissed by the reader. I can think of a set of simplifying assumptions that might lead to this prediction (e.g., (i) plate motion relative to the mantle is negligible, the change in the orientation of the principal axes of non-hydrostatic moment of inertia is instantaneous, (iii) those subsequently do not change at all), but that's the author's job, not the reader's. This erroneous assertion is repeated several times further below in the manuscript.

R: We echo the reviewer in our revised text to make our statement more justified (lines 230-240):

"As defined as the migration of the maximum moment of inertia (I_{max}) to align with Earth's spin axis, TPW occurs as a rotation about an Euler pole controlled by the minimum moment of inertia (I_{min}) that is equatorial and is therefore predicted to circumscribe a great-circle APW path. Identifying TPW as a great-circle APW path also assumes that plate motion of the continent relative to the mantle is negligible, the change in the orientation of the principal axes of non-hydrostatic moment of inertia is instantaneous, and those subsequently do not change at all. The similar amplitude and synchronicity of these 5 continents indicate their individual plate motions are negligible relative to the shared TPW motion. Numerical simulations indicate such a change in the orientation of the principal axes of non-hydrostatic moment of inertia can be completed within 10 Ma."

Q28: Lines 202-203: "These systematic gaps in the APW paths of all continents are consistent with the stroboscopic effect expected for TPW...". This is another assertion made, but backed by no evidence. If the authors believe this is true, they should quantify it. What are the odds that they've captured directions 60° apart separated by 10 Ma, but not captured any pole positions during that 10 Ma on 5 different continents? My guess is that the systematic gaps are not consistent at all. It's up to the authors to present evidence that they are consistent and to quantify the argument probabilistically.

R: In order to quantify the very low probability of capturing TPW "in the act", we have refined our main text explanation of this statistical matter, added a Figure 4, and a Methods section:

New Figure 4:

Fig. 4. Probability of sampling true polar wander. **a** True polar wander (TPW) angle as a function of time with an initial condition of 25°. **b** TPW speed (black line) and probability function (shaded gray) as a function of time.

New Method section (lines 550-586):

“Probability of sampling true polar wander. For any quantity, φ , that varies as a function of time, the probability density function (PDF) describes the relative amount of time spent at each value of φ or, alternatively, the likelihood that a particular value of φ will be sampled. For TPW, we let $\varphi(t)$ be the angular separation of the rotation axis relative to a fixed geographic axis (TPW angle), or the angular distance from one estimate of the location of the spin axis to some later estimate. An un-normalized PDF for $\varphi(t)$ is then given by

$$f(\varphi) \equiv \left| \frac{dt}{d\varphi} \right| = \frac{1}{|d\varphi/dt|}, \quad (1)$$

so that the probability of sampling φ within a given angular interval is

$$P(\varphi_1 < \varphi < \varphi_2) = \alpha \int_{\varphi_1}^{\varphi_2} f(\varphi) d\varphi, \quad (2)$$

where α is a constant chosen so that the total probability $P(0^\circ < \varphi < 90^\circ)$ equals 1. Equation (1) shows that the likelihood that φ is sampled is exactly inversely proportional to the TPW speed, $|d\varphi/dt|$. This straightforward relationship between TPW sampling probability (PDF) and TPW speed shows that faster instantaneous TPW speeds are prone to be undersampled. Technically speaking, the uphill battle to observe rapid TPW is nothing more than the age-old stroboscopic effect, or aliasing: the difficulty of studying rotating planets, reciprocating blades, oscillating fans, or vibrating strings with discrete data; temporal undersampling is a major hurdle for understanding geologic processes and TPW is not an exception.

We now demonstrate an example of the undersampling problem with a simple modeled TPW excursion. In the following example, we use the simple analytical framework of Tsai and Stevenson to describe the TPW due to a chosen moment of inertia tensor variation. In this formulation, the Liouville equation for a viscoelastic planet is analytically solved for a given perturbation of the moment of inertia tensor, following Munk and MacDonald, to obtain the TPW angle. For simplicity, we chose a moment of inertia variation that is sinusoidal with a period of 150 Myr, with an average viscosity of 3×10^{22} Pa s, and an amplitude of $10^{-5}C$ (where C is the Earth's moment of inertia). This variation is chosen to very roughly approximate the observed TPW described in the next section. Equation (12) of Tsai and Stevenson then yields the TPW angle as a function of time, $\varphi(t)$, which is plotted in Figure 4a for an arbitrary chosen initial condition. The associated TPW speed (black line) and the PDF (shaded gray) for this TPW curve are plotted as a function of time in Figure 4b. As shown, the maximum instantaneous TPW speed is about $2.2^\circ \text{ Myr}^{-1}$ (24 cm yr^{-1}) and is associated with a minimum in the PDF, showing that it is the least probable value to be observed if one randomly samples the distribution. One can also compare the probability of sampling within a finite range by using Equation (2), or by simply reading time intervals from Figure 4a. For example, the values of φ in the range $4^\circ < \varphi < 9^\circ$ represent approximately 32% of all measurements whereas a similar 5° range $40^\circ < \varphi < 45^\circ$ represents only about 1.5% of all measurements. For this example, then, it is about 20 times more likely to sample within the first range of φ (low instantaneous TPW speed) compared with the second range (maximum instantaneous TPW speed)."

Q29: Line 212: Any complete discussion of mass anomalies should include the waxing and waning of ice sheets.

R: We added an additional whole paragraph in order to talk about the effects of glaciation on TPW (lines: 285-305):

"It is also possible that the waxing and waning of ice sheets across Gondwana contributed to the mass anomalies driving O–S TPW. In particular, there is a migration of glacial centers from northern Africa to southern Africa–South America, where glacial and periglacial strata in the former region are predominantly Ordovician and those in the latter neighboring regions are predominantly latest Ordovician or Silurian. That is, the mass load associated with incipient Ordovician glaciation applied in northern Africa could have been driven to the equator by TPW, causing southern Africa–South America to move to the pole and thus moving the glacial center there in the earliest Silurian (Fig. 5). This hypothesis, by extension, would also predict ensuing oscillatory Silurian–Devonian TPW back in the direction of northern Africa (in order to drive the glacial center in southern Africa–South America equatorward), which has indeed been previously hypothesized, but the assessment of which is beyond the scope of our study on O–S TPW. In the Cenozoic, however, glaciation is typically regarded more as an effect of TPW rather than a cause of it, as the amplitude of glacially-induced TPW is smaller than TPW driven by mass reorganizations in the mantle. Nevertheless, given the larger size of the Palaeozoic pan-Gondwanan ice sheet, and thus its presumably larger mass load, glacial loading deserves further investigation for potentially driving the O–S TPW event. If valid, such an interpretation—the incipient glacial load causing TPW, which then led to more severe

glaciation as Gondwana became centered over the South Pole—presents a fascinating potential feedback between TPW and glaciation.”

Q30: Line 265: It’s my understanding that relative paleo-longitude can be determined this way, but not absolute paleo-longitude. This should be clarified in the manuscript.

R: We now clarify the constraint is relative palaeolongitude.

2. Responses to Reviewer #2

Q1: As no radiochronometric data are available, the ages of these units are only estimated using a poorly biostratigraphical control and some uncertainties exist whether the age of the Rongxi Fm. is Telychian or Aeronian (Rong et al., 2019). Based on available data, the Telychian age used in this paper seems reasonable (Huang et al., 2017) and will need to be confirmed in the future.

R: Actually, new paleontological data from Yangtze region indicate that the Rongxi Fm in our and previous studies' sampling area is early Telychian. See these papers:

1. Chen, Z., Männik, P., Fan, J. *et al.* Age of the Silurian Lower Red Beds in South China: Stratigraphical Evidence from the Sanbaiti Section. *J. Earth Sci.* **32**, 524–533 (2021). <https://doi.org/10.1007/s12583-020-1350-6>
2. Zong R W, Liu Y L, Huang L B, et al. Trilobites from the Silurian “Lower Red Beds” of Wuhan, South China: stratigraphic and paleogeographic implications[J]. *Palaeoworld*, 2022, 31(2): 239-248. <https://doi.org/10.1016/j.palwor.2021.05.004>

Q2: (1) Based on only 6 sites, the Huixingshao pole does not meet this criterion and the minimum of 8 sites required to calculate a paleomagnetic pole. In addition, the precision parameter (K) for the VGP distribution is ~90.8 (not shown in the Table 1 supp. data), which could indicate some suspicion of inadequate averaging of secular variation (>70; (Meert et al., 2020)). As this issue was not discussed in the paper, I strongly suggest to the authors to provide the Deenen et al. (2011) confidence bounds.

(2) The recalculated ~438 Ma Rongxi pole fulfills the R2-criterion but no details were performed by the authors about the used sites of Opdyke et al. (1987) and Huang et al. (2000): Line 123-125: the recalculated Rongxi pole was calculated using the the Tongzi, Dagan, and Shiqian sections of Opdyke et al. (1987) and Huang et al. (2000). This was repeated in Line 38-39 of the supplementary data and the name SDT was used in the Figures for this pole (Shiqian, Dagan, Tongzi). In the Supplementary Fig. 5b, the site-mean direction for the Rongxi Fm. is illustrated, and particularly the C1 component of Huang et al. (2000), considered as primary. The results and the Fisher 's statistics are indicated in orange. I was able to check and I get exactly the same values and the same stereo using:

- the sites 8,9,10,11,12,13,14 from Shiqian (Huang et al., 2000)

- the site a from Tongzi (Opdyke et al., 1987)

- the sites J, K,L,M,N,O,P,Q,R,S,T,V from the Xiushan section (Opdyke et al., 1987)

The associated paleomagnetic pole with this mean direction is located at 10.3°N, 196°E.

This Rongxi pole is different from the pole used in Table 2 (-17.9°N, 20.3°E). Although I was not able to obtain exactly these values, I obtained a close result by using the sites of Shiqian, Dagan, and one site of Tongzi (i.e., the SDT acronym) ...

Therefore, I don't understand why to use in the supplementary Fig. 5b the Xushian sites and not the Dagan sites and finally in the recalculated Rongxi pole of Table 2 the Dagan sites. In addition, the Dagan sites were considered as C2 component by Huang et al. (2000) and interpreted as a "remagnetization" component (see the caption of the Fig.5b in the supplementary data). If you have decided to use the Dagan sites as primary in the calculation of your new Rongxi pole (SDT), then you must absolutely justify this choice.

In short, the calculation of your new ~438 Ma Rongxi paleopole is terribly confusing and only a table can help to clarify it!

R: As the reviewer pointed out, the K value and number of sites don't fulfill the Reliability criterion #2 of Meert et al. (2020), which "could indicate some suspicion of inadequate averaging of secular variation". According to Meert et al. (2020) Reliability criterion #2, a test for sufficiently averaging palaeosecular variation (PSV) is $10 \leq K \leq 70$. Indeed, when we calculate the K value for the previous data of Huang et al. (2000) and Opdyke et al. (1987), both studies have K values > 70 (new Supplementary Data 1). However, by combining all Rongxi Fm and Huixingshao Fm data of our study (28 sites) and the previous studies, we get a $K = 48$ that is squarely within the ideal range for averaging PSV. This combined results also passes a fold test.

We rewrite these parts related to the data analysis:

In main text (lines 125-142):

"However, the K value of dispersion of the virtual geomagnetic poles (VGPs) of these six sites is 90.3 (Supplementary Data 1), exceeding 70, which suggests that these data may not average out the palaeosecular variation (PSV). To overcome this issue, we sought to combine our new data with the most reliable coeval previous data.

We reassign the ages of existing Silurian palaeomagnetic results according to a recently updated stratigraphic timescale (Supplementary Fig. 1). A notable revision in these age reassignments is that the Rongxi Formation previously regarded as ca. 420 Ma in age is in fact early Telychian (ca. 438.5–437 Ma) (Supplementary Fig. 1). Again, data from these previous studies seem not to average out PSV (Supplementary Data 1; detailed analysis in supplementary text). Nonetheless, after combining all data from the Rongxi and Huixingshao formations (total 28 sites), a K value of 48.4 is achieved, which is below 70 and suggests sufficient averaging of PSV. Furthermore, these data also pass a fold test at 99% confidence (k in geographic coordinates is 7.64, in stratigraphic coordinates is 31.17). This new early Silurian pole (S_1M) calculated by averaging the VGPs from the Rongxi and Huixingshao formations plots far from all younger poles and earns a reliability index of 6 of 7 (see supplementary text for details)."

In supplementary text (lines 43-87):

“The K value of dispersion for the mean virtual geomagnetic pole (VGP) of the SY sites is 90.3 (Supplementary Data 1), which is >70 and suggests it may not average out the palaeosecular variation (PSV). We do note that if the sedimentary data are treated at the sample level instead of the site level, the K value for 101 samples is 65.5, where both these values satisfy the requirements of Reliability criterion #2 of Meert et al. concerning the averaging of PSV. Nonetheless, the A95 (1.81) resulting from the sample-level mean VGP is slightly below the A95min (1.95) of Deenen et al., again suggesting that PSV may be slightly under-sampled by the SY section. We next sought to combine our results from the SY section with previous results from coeval strata in order to adequately average PSV.

Previous studies mixed results from the Rongxi Formation (Fm), the Huixingshao Fm, and the Xiaoxi Fm of different regions together, which reduces the temporal resolution of the Silurian palaeopoles. Therefore, we reanalyze those paleomagnetic data of the Rongxi Fm of Huang et al. and Opdyke et al. (Supplementary Data 1). After reviewing the papers of Huang et al. and Opdyke et al., we note that data from the Rongxi section, Xiushan county of Opdyke et al. are both from the Rongxi and Huixingshao formations (Supplementary Fig. 1), which cannot be separated from each other. The K value of VGPs of these 12 sites is 121.6 (Supplementary Data 1), which suggests they did not average out the secular variation either. The two Guandi Fm data from Qujing (Supplementary Fig. 1) were thought to be contemporary with the Huixingshao Fm. However, new palaeontologic studies suggest that the Huixingshao Fm belongs to the middle Telychian and the Guandi Fm covers the Ludlow Epoch. Specifically, the red beds of the Guandi Fm (Guandi II and Guandi III Members) are of late Gorstian to early Ludfordian Age (Supplementary Fig.1). Therefore, these two data have a distinct age from the data from the Rongxi and Huixingshao formations, hence we reject them for our combined calculation. Besides that, data of sites 6 and 7 from the Shiqian section are from the Huixingshao Fm (Supplementary Fig. 1). The remaining data including sites 8–14 from the Shiqian section and site A from the Songkan section, Tongzi, both of which clearly belong to the Rongxi Fm. The K value of these 8 sites is 133.7 (Supplementary Data 1), which still exceeds 70. Even if we combined all the data from the Rongxi and Huixingshao formations of Huang et al. and Opdyke et al., comprising 22 sites, the K value of 87.6 still exceeds 70 (Supplementary Data 1). These results suggest that all the previous data cannot average PSV, and hence are unreliable on their own. Nonetheless, considering that the Rongxi and Huixingshao formations span only ~ 4 Myr, we combine our Huixingshao Fm data with all these data from the Rongxi and Huixingshao formations of Huang et al. and Opdyke et al., totaling 28 sites. The mean of the combined VGPs has a $K = 48.4$ (Supplementary Data 1), which is squarely in the acceptable range of 10–70 and suggests PSV is adequately averaged in the new combined result. These data also pass a fold test (“ $Xi1$ ” test) at 99% confidence ($Xi1 = 5.956 < Xi1_{critical} = 8.703$). The mean pole of these VGPs (S_{1M}) is located at $6.8^{\circ}N$, $195.6^{\circ}E$, with $A95 = 4$ (Fig. 1f). This S_{1M} pole fulfills 6 of 7 reliability criteria proposed by Meert et al., and hence is a very reliable palaeomagnetic pole.”

Q3: In addition to the supplementary Fig.3, stereographic projections and magnetization intensity decay curves for some examples will be useful to see the behavior of these samples. It is important to show these curves because you mention the high unblocking temperature at the Line 108 (main text) and also in the Lines 53-54 (supplementary data) to justify the hematite

as the main remanent carrier. Fig.8 in the supplementary data is welcome. Since the authors suggest that the SK and ST sections are remagnetized, it would be important to show whether there are differences in magnetic mineralogy between these sections.

R: In the revision, we added Fig.1a,b,c and Supplementary Figs. 3 and 4 to demonstrate the demagnetization characteristics of our data. All have stereographic projections and magnetization intensity decay curves.

Q4: (1) Unfortunately, no fold test is available for the Huixingshao pole although 3 different sections were sampled from each side of the syncline. Beyond the small number of sites used, this does not allow to consider this pole as robust and therefore the question arises if it can be used to characterize the position of the South China Block.

(2) As outlined above, the calculation of the newly Rongxi pole must be reviewed, and it is important to point out if the chosen sites pass a fold test.

R: Addressed in response to Q2 above. We reanalysed the data, and the new result now includes data from both the Rongxi Fm and Huixingshao Fm, and it passes a fold test at 99% confidence.

Q5: Using paleomagnetic comparison between the Rongxi data from the Xiushan section (Opdyke et al.,1987) and the Rongxi data from the Shiqian/Daguan/Tongzi (SDT pole), the authors suggest a $12.1 \pm 3.9^\circ$ clockwise rotation (Fig.7. supplementary data). First, I cannot check this value without to be sure of the used sites for the SDT pole as previously discussed.

One question I have is why use the Daguan sites (C2 component) in the calculation? It's interesting to see that these are the sites that pull the two directions apart. Moreover, the Daguan sites are geographically very distant which cannot exclude a complex tectonic history between these areas? Huang et al. (2000) have seen that the directions (C1) of Shiqian were similar to those of Xushian and this was an important argument that allowed him to suggest that the C1 component was indeed primary! According to this statement there was no need to put a rotation of 12° between the sites of Xiushan and Shiqian for the Rongxi Fm as it is proposed in this paper.

In addition, the authors use this rotation to correct Huixingshao Fm pole of the Yongdong section (SY). This would correspond to a different history and a very local rotation (evidence of that?). Perhaps the fault marked in Fig. 1 but all of this remains highly speculative. Another comment about this rotation is that the authors claim that the SK and ST section are remagnetized, contrary to the SY section. But by looking in detail, the SK-ST direction share a common direction with the SDT direction (McFadden and McElhinny, 1990). It is not clear why these sites (SK-ST) are excluded contrary to the Yongong (SY) section.

Therefore, is the rotation proposed by the authors not an artifact related to the choice of pmag sites used and not a tectonic problem? I ask this question and I think that additional data should be provided (structural data, AMS, fold test to prove which limb are remagnetized...). In view of the elements, neither of the two poles can validate this R5 criterion.

R: Addressed in response to Q2 above. The difference between data from Xiushan section and Shiqian section may be just inadequate averaging of the secular variation. Either way, we deleted this part.

And, yes, the data from Daguang (C2) are all remagnetized, therefore we exclude them from our analysis this time.

Q6: Lines 111-113: As illustrated in Fig.5a (supplementary data), two antipodal directions were revealed for the Huixingshao Fm with a positive reversal test (C-class) according to McFadden and McElhinny (1990). This argument is the only one left to justify the primary character of the high temperature component since there is no fold test. It should be remembered that a remagnetization with two-polarities is also possible. Interesting, the authors provide a magnetostratigraphy of the Yongdong section for the Huixingshao Fm. (Fig. 4 supplementary data). Clearly, we can see the reversal state of the SY8 site. The second reversed site is the SY10 as illustrated in the Fig.5a and listed in the Table 1 (supplementary data). For the SY10 site, we cannot clearly see a reversion in the declination and inclination curves! Besides, the authors used for this site the expression “normal with reversal samples”. In the inclination curve I can see only one specimen in the reversed polarity. This seems to be in total contradiction with the Table 1 where I can read 20 specimens showing a site-mean inclination of -39.6° ... I therefore ask the question if there was not only one specimen in reverse polarity and that the authors have changed the polarity of all other specimens in normal polarity. If this is the case, then there would be only one zone of true reverse polarity in the section SY and the reversal test would be inconclusive. We need to be sure about that and the magnetostratigraphy needs to be coherent with the site-mean directions.

R: Yes, all data from site SY8 are of reversed polarity. Only two samples are reversed at site SY10, as showed in Fig. 2. Because South China was located close to the Equator at that time, we use declination, instead of inclination, to determine whether data are reversed or normal.

In this reversion, we keep SY10 as normal polarity, and our data fail the reversal test.

Q7: Fig.6 supplementary data show that the SDT pole and the Huixingshao pole are different from younger poles to fulfill this R7-criterion.

In summary, we can assign the R-criteria 4 and 5 for the Huixingshao and Rongxi poles. This differs from the 7 and 6 rating using the Van der Voo's criteria (see the Table 2). This highlights that objectively, the recalculation of the Rongxi pole is clearly more robust than the Huixingshao pole from the Yongdong section (SY) if the Rongxi passes a fold test. For the proposed TPW hypothesis and analysis to be convincing it is therefore necessary to insist on the use of the most robust poles, which is not the case with the Huixingshao pole.

R: Addressed in response to Q2 above. The new calculated pole fulfills 6 of 7 Reliability criteria (Meert et al., 2020) (Supplementary Data 2), only failing the reversal test (which is obviously subject to some luck of the draw depending on the frequency of reversal at a given time period, which can vary significantly).

Q8: Lines 65-66: “Ediacaran–Cambrian periods, and this interval has been proposed to have experienced large-scale (60–90°) true polar wander^{26–32}.”

For a complete bibliography of this topic, small TPW oscillations were also recently proposed by Antonio et al. (2021) during the Cambrian times (<535 Ma) that follow these large TPWs. This indeed underlines that this is a turbulent period from the paleomagnetic point of view and a further argument that this period may extend into the Silurian as proposed in this paper. The reference “27. Kirschvink, J. L. & Raub, T. D. A methane fuse for the Cambrian explosion....” will be more adapted in the discussion about the link TPW-biodiversity (Lines 355-393).

R: Great suggestions. We have added this reference (Antonio et al., 2021) in this description. Reference to Kirschvink and Raub (2003) has also been added in line 413.

Q9: Lines 67-68: “the movement of the entire solid Earth (mantle and crust) relative to the liquid core in order to stabilize Earth rotation”

As the liquid core is turbulent (with convection movements > ~km/yr), it should be better to use the reference for the movement of the Solid Earth the spin axis (fixed into the celestial reference frame).

R: We rewrite it as “True polar wander (TPW) is the movement of the entire solid Earth (mantle and crust) relative to Earth’s spin axis in order to stabilize Earth rotation.”

Q10: Line 78: “However, no follow-up research has reexamined this TPW event”. When we mention a plausible Ordovician-Devonian TPW, I suggest adding the reference of Piper (2006) who also published on this topic 10 years later than Van der Voo (1994). Therefore, your line 78 is inappropriate.

R: We rewrite this sentence to address the study of Piper et al. (2006) (lines 86-88): “Although Piper et al. revisited this TPW interval, they only studied the late Silurian–early Devonian part”

Q11: Line 165-193: “palaeomagnetic poles from Laurentia during this time are characterized, in contrast, by much less APW, and almost essentially a standstill”. This discussion about the motion of the Laurentia is convincing! Intriguing, Australian poles show a similar behavior during the Cambrian (<540 Ma). As Australia is in opposite position along the great circle (Fig. 2), this could suggest a fixed Imin axis position in the Early Paleozoic times.

R: Yes, they should be. Details can find in Mithcell et al. (2012) and Evans et al. (2003)

Q12: Lines 244-: About the paleogeographic reconstructions based on TPW. The available paleomagnetic data for this interval exhibit abrupt directional variations, and previous efforts have been made to explain this erratic behaviour (appropriately recognized and cited in the paper). Their findings show that global APW paths imply high velocities incompatible with plate tectonics, so they appeal directly the hypothesis of large TPW events

to account for the misfit. It is worth noting that this paper is missing a few sentences about the other possibilities invoked in the literature to explain these movements (collision, non-uniform Earth's magnetic field) and why the authors feel that TPW is the most appropriate solution?

R: We appreciate this wise suggestion. In the revision, we add some sentences to talk about other mechanisms which can cause anomalous polar wander. We now discuss the possible effects of tectonically-induced vertical-axis rotation (potentially affecting declination) and non-uniformitarian magnetic fields (potentially affecting inclination), both of which cannot comprehensively explain the global palaeomagnetic data. Meanwhile, TPW provides a sufficient and simple explanation of the data. This revised text is as follows (lines 147-162):

“During this time interval, South China experienced a region tectonic movement, however it was restricted to only its southeastern part (Cathaysia terrane). Our early Silurian data and the late Ordovician data are from northwestern South China (upper Yangtze terrane), which was largely unaffected by this tectonism. In addition, the regional tectonism should have only induced large differences in the declination of these data (due to potential vertical-axis rotation), but cannot explain the large inclination difference that is observed corresponding to a $\sim 28.5^\circ$ change in palaeolatitude. Non-uniformitarian magnetic fields (e.g., quadrupolar or octupolar) may also result in anomalous polar wander. However, both quadrupolar and octupolar fields would cause palaeolatitude to *increase*, while our Silurian data demonstrate a lower palaeolatitude (9.2°) than in the late Ordovician (-19.3°). Furthermore, both non-dipole cases would only affect inclination and therefore cannot explain the even larger anomaly in terms of the $\sim 59^\circ$ declination change. Lastly, an oscillation between polar and equatorial dipoles (if possible on Earth) could affect declination, but would predict a $\sim 90^\circ$ change that is not observed.”

Q13: Fig. 1c represents the Gondwana paleolatitude and allows to delimit the tightly timing of the TPW 450- 440 Ma. To confirm such global timing, it should be added on the same Figure the curves of the 6 other continents.

R: Excellent suggestion as a nice illustration of global timing. We add the arc distance curves for the other 5 continents in Fig. 1c.

Q14: This calculation was based on the Wu et al. (2020)'s compilation for Baltica/Laurentia and Gondwana. It is important to note that the Wu's approach to calculate the APW paths (weighted running mean poles) is different from a simple comparison of paleopoles as it is the case in this paper for the South China block. Such differences could provide different results in the APW paths, and it is important to compare the different methods. For example, the APW path for the Gondwana of Torsvik et al. (2012) was calculate by running mean path and a 20 Ma sliding windows. Using these models, the timing of the TPW will surely be different. A comparison of the different methods could be provided in the supplementary data.

R: Good point. We added text to discuss the different methods and results (lines 173-187):

“Data from Baltica and Gondwana represent recent synthetic APW paths, which consider the age error and the quality of the data. For comparison, we also calculate the arc distances for Baltica and Gondwana from 450–430 Ma using the synthetic APW paths of Torsvik et al. (Supplementary Data 3), which are $51.2^\circ \pm 8.2^\circ$ and $24.5^\circ \pm 18^\circ$, respectively. While the results for Baltica agree with both methods, the large difference of the two synthetic APW paths for Gondwana reflect either the larger 20 Myr age bins of Torsvik et al. oversmoothing the data and/or the lack of poles during this time interval which is non-ideal for synthetic methods. Nonetheless, at least four continents demonstrate similar large amplitude and synchronous polar motion. As discussed, regional tectonics and non-uniformitarian geomagnetic fields cannot explain this systematic global APW anomaly. Plate motion, driven by slab subduction and mantle convection, also cannot explain these synchronous and similar large amplitude movements of multiple isolated continents either, as it requires relative motion between different plates with different velocities (speeds and/or directions).”

Q15: Line 321: “reconstruction of the latest Katian, Niger”

Double space between of and the.

R: Redundant space deleted.

Q16: Line 431-432: The sampling part is not detailed enough and must be developed in the supplementary data.

- So that readers not used to the local geology can follow the text, it is necessary to improve some figures :

o The supplementary Fig.1 needs to be reviewed. In this Figure, we have to see the locations of the different sections cited in the text (Shiqian, Dagan ...). A simplified regional tectonic setting between these sections is important because the authors used these sections for paleomagnetic corrections.

o The acronyms used need also need to be judiciously chosen explicated in the caption to help the reader: for example the Yongdong village is SY.. it will be better to use the Section Yongdong to easily understand the acronyms.

R: We add a map in Supplementary Figure 1 that shows the locations of the different sections. For the acronyms, we change them as the reviewer suggested (e.g., Section Yongdong-SY)

Q17: Line 457: Be careful, the manufacturer of the KLY4 is AGICO and not ASC Scientific (it is a reseller).

R: We change it to AGICO.

Q18: Lines 463-464: I think that Paleomac was also used for the Figures and calculations (see the Fig. 5), not only the reversal test.

R: We move this to the supplementary text and rewrite this sentence accordingly (lines 18-19):

“PaleoMac was used to conduct the fold and reversal test and produce Fig. 1 and Supplementary Figs. 3, 4, 5.”

Q19: As I just said the geological context figure must be improved. The location of the different sections will allow to easily locate the other sections mentioned in Fig.2.

R: We add a map in Supplementary Figure 1 that shows the locations of the different sections.

Q20: Fig.3: Add stereographic projections and magnetization intensity decay curves for some examples to reinforce your paleomagnetic dataset.

R: We redrew these demagnetization characteristic figures. Three new figures have been added: Fig. 1 and Supplementary Figs. 3 and 4. All samples illustrated include both stereographic projections and magnetization intensity decay curves.

Q21: Fig.4: Explain the misfit of the SY10 site (only one reversed specimen while the site is considered as reverse polarity in Table 1). In addition be careful to the typo errors (declination/inclination/meters are badly written).

R: Firstly, because the inclination is low, which suggests South China should be located close to the Equator, we therefore deem it better to determine the polarity of data by its declination. In our new Supplementary Table 1, we consider site SY10 as normal polarity now (only a couple samples are reversed). Typo errors have been corrected in new Fig. 2.

Q22: Fig.5: Add a stereo for the calculation of your newly Rongxi paleopole that you will use in the paper and justify the used sites (e.g., from Dagan). It is missing the number of sites used in the calculations (N) in your Figure.

R: We rewrite the part of the data analysis—check response to Q2—and delete this figure.

Q23: Fig.6: We need a reference in the caption for the poles of reference in red.

R: This figure now is Fig.1f and Supplementary Fig. 5, and a reference has been added in the caption.

Q24: Fig.7: this figure needs to be coherent with the calculation of the Fig.5. Use different colors for different locations.

R: As we reanalyse the data, this figure is not suitable now. We deleted it.

Q25: Table 1: add columns for P.lat and P.lon rather than lines as you did. This will save space and the table will be clearer. Add the sites in the table (or a new) for your Rongxi direction, the associated mean directions and the mean pole. Add also the Huixingshao pole before and after rotation!

R: We replace the previous Table 1 with a new Supplementary Data 1. Columns for Plat and Plong are added to the Table. Data of Huang et al. (2000) and Opdyke et al. (1987) are also added. As we reanalyse these data, the Huixingshao Fm pole before and after rotation is not needed now.

Q26: Table 2: Check the reference for the Tarim craton (Huang et al.1). Use the R-criteria (Meert et al., 2020) instead of the Q of Van der Voo. Add also if they are paleomagnetic poles or synthetic poles from averages (+ the method of calculation).

R: We change the reference for data of Tarim in new Supplementary Data 2, added columns of R/Rf, Type and K, to evaluate the data.

3. Responses to Reviewer #3

Q1: There is no actual data in the main text of the paper, making the raw data difficult to examine or scrutinise, which are the basis of all the later arguments. If permitted by the number of figures allowed, please add at least one of figures S3, S4, or S5 into the main text.

R: We added Figure 1 and 2 to the main text of the paper to illustrate the raw data.

Q2: None of the axes in Fig S3 have values on them, and only the vertical axis has a label. Please add values and labels.

R: In the new Fig. 1 and Supplementary Figs. 3 and 4, we added the axes scales and labels.

Q3: The orthogonal project plots in Fig 3S are said to be ‘representative’ in the caption, but there are no data from SY11, SY12, or SY13 — this means there are no raw data from these sites in the paper at all. Moreover, SY10-1B and SY10-14B are included in the orthogonal projection plots despite not appearing to be representative of SY10 as a whole. Further discussion on these two datasets would be appreciated. For instance, they have very different inclinations, so it would be good to see more clarity on which data went into calculating the average datapoint for the SY10 point in Fig 4a that is similarly orientated to SY8. With this in mind, is it fair to call them reversed samples, given only one of them has an orientation close to the points from SY8.

R: We add demagnetization diagrams of representative samples from sites SY 11, 12, and 13 in new Fig. 2 and Supplementary Figure 3.

As all samples are illustrated in geographic coordinates, the high temperature component may differ from each other. Data are tilt corrected to be used as reliable primary magnetization directions. All these data are used to calculate the site mean direction of SY10.

Q4: A lot of the data presented is limited to final, average directions. While this is the key information that enables the plate reconstructions to be conducted, it would be more transparent to include all of the data in Fig S4 as well as the averages. Personally, I like to see this in a paper, as it enables readers to assess the data for themselves.

R: We move the figure in question to the main text as Figure 2 now. Every data point in the figure represents specimen level data, not average directions.

Q5: Finally, the low-temperature component is said to be a VRM consistent with the direction of the present-day local geomagnetic field. While the LT component in several of the ST and SK samples are to have similar orientations, this doesn't appear to be the case for SY10-1B and SY10-14B, or SK2-2. Perhaps an equivalent of Fig S5 for the LT component with the present-day local field orientation would help to demonstrate how many samples do carry an LT component with the same direction as the present day field. This would hopefully help build confidence in the data.

R: We add a new Supplementary Fig. 7 to demonstrate the low temperature (LT) component, and its closeness to the present local field

References:

Landing, E., and MacGabhann, B.A., 2010, First evidence for Cambrian glaciation provided by sections in Avalonian New Brunswick and Ireland: Additional data for Avalon–Gondwana separation by the earliest Palaeozoic: *Palaeogeography, Palaeoclimatology, Palaeoecology*, v. 285, p. 174–185, doi:10.1016/j.palaeo.2009.11.009.

Meert, J.G., Pivarunas, A.F., Evans, D.A.D., Pisarevsky, S.A., Pesonen, L.J., Li, Z.-X., Elming, S.-Å., Miller, S.R., Zhang, S., and Salminen, J.M., 2020, The magnificent seven: A proposal for modest revision of the quality index: *Tectonophysics*, v. 790, p. 228549, doi:10.1016/j.tecto.2020.228549.

Swanson-Hysell, N.L., Ramezani, J., Fairchild, L.M., and Rose, I.R., 2019, Failed rifting and fast drifting: Midcontinent Rift development, Laurentia's rapid motion and the driver of Grenvillian orogenesis: *GSA Bulletin*, v. 131, p. 913–940, doi:10.1130/B31944.1.

Reviewer #1 (Remarks to the Author):

The authors responded in good faith to my comments and suggestions for an earlier version of the manuscript. These responses have greatly strengthened the manuscript in my opinion. For this version of the manuscript I have only two comments/suggestions that probably require a very minor revision of the manuscript.

(1) In my prior review, I pointed out that "Late Silurian" and "Early Silurian" should be written as "late Silurian" and "early Silurian" as, according to the geologic time scale I keep on my desktop, these are not formally recognized divisions of geologic time. The authors adopted my suggestion. So far, so good. It appears, however, that they have changed all uses of "Late" and "Early" to "late" and "early". This is a mistake if applied to formally recognized divisions of time such as "Early Ordovician" and "Late Ordovician". These errors could probably be fixed in the galley proofs and not necessarily require another round of manuscript revision.

(2) On lines 156-157 the authors write "...both quadrupolar and octupolar fields would cause palaeolatitude to increase, while our Silurian data demonstrate a lower palaeolatitude (9.2°) than in the late Ordovician (-19.3°)...". I do not understand how this assertion can be true. Both quadrupolar components and octupolar components can cause paleolatitudes inferred assuming a purely dipolar field to either increase or decrease. It all depends on the sign of the component. Maybe the authors are assuming these components only come in one sign? If so, I don't think there is any justification for that. I think that this is an error in reasoning that needs to be corrected.

Minor comment: Please insert "of" after "case" in line 74.

Richard G. Gordon

Reviewer #2 (Remarks to the Author):

The authors have succeeded in solving all the problems identified in my previous review concerning the quality and robustness of the paleomagnetic data presented. They combined different dataset to present new calculations about the Devonian Rongxi and Huicingshao Fm. Consequently, they obtained in this new version a positive fold test for the data, which give more confidence on these results. As discussed, these data do not fulfill a reversal test. In addition, the selection of study and used sites in the calculation is now coherent as showed in the different Figures. I also appreciate the Figure 1 of the supplementary material to see the location of different studies. Calculation of the TPW is consistent and the new Figure 3 shows a global trend for most of continents (except Laurentia), which what in the state of current knowledge and the poor-dataset can be accepted. A significant addition in this new version is the paragraph on the probability of sampling true polar wander.

Some minor comments:

Figure 5: The name or the age of paleomagnetic poles in the Figure are not indicated. This makes the kinematics in the Figure difficult to analyze since you have chosen to move the Mollweide projection and not use a fixed projection. You re-used the panel of colors of your Figure 3, but the continents are not in the same color. Please add in the Figure caption the link with the Figure 3 and the motion of the Mollweide projection, which can be confusing for many readers. Since you added the evidence of glaciations on the Figure, do you have any evidence of warm climate to add?

Line 14-16 in the supplementary material: the site-mean directions is not calculated by means of Fisher statistics. This is this is a common mistake that is spread in many papers. If you use Bingham statistics (for example) the site-mean directions will be the same. You can-rewrite your sentence like this: "The remanence directions were analyzed using least-squares analysis or great circle intersections as employed by the Paleomagnetism.org web application, site-mean directions were calculated by averaging the different sites, and the statistical parameters were calculated assuming a Fisherian distribution".

Reviewer #1.

Question 1: In my prior review, I pointed out that "Late Silurian" and "Early Silurian" should be written as "late Silurian" and "early Silurian" as, according to the geologic time scale I keep on my desktop, these are not formally recognized divisions of geologic time. The authors adopted my suggestion. So far, so good. It appears, however, that they have changed all uses of "Late" and "Early" to "late" and "early". This is a mistake if applied to formally recognized divisions of time such as "Early Ordovician" and "Late Ordovician". These errors could probably be fixed in the galley proofs and not necessarily require another round of manuscript revision.

Response: In this revision, we only make lowercase the "late" and "early" before "Silurian". For other formal intervals, we retain uppercase, like "Late Ordovician".

Question 2: On lines 156-157 the authors write "...both quadrupolar and octupolar fields would cause palaeolatitude to increase, while our Silurian data demonstrate a lower palaeolatitude (9.2°) than in the Late Ordovician (-19.3°)....". I do not understand how this assertion can be true. Both quadrupolar components and octupolar components can cause paleolatitudes inferred assuming a purely dipolar field to either increase or decrease. It all depends on the sign of the component. Maybe the authors are assuming these components only come in one sign? If so, I don't think there is any justification for that. I think that this is an error in reasoning that needs to be corrected.

Response: Yes, it is true that both quadrupolar components and octupolar components can cause paleolatitudes to either increase or decrease depending on if they are of reverse or normal polarity, respectively. We have revised the text to explain how, gaming out either sign of the non-dipole field, that the data cannot be explained by such non-uniformitarian fields (Lines 159-164):

“Non-uniformitarian magnetic fields (e.g., quadrupolar or octupolar) may also result in anomalous apparent changes in latitude³⁴. However, in order to explain the reduced inclination of the Late Ordovician data (35°) to our Silurian data (18°), one would have to claim a same-sign octupole that was stronger than 20%, which is more extreme than any previous claims in the Phanerozoic¹, and an opposite-sign octupole would increase, not decrease, inclination.”

Reviewer #2.

Question 1: Figure 5: The name or the age of paleomagnetic poles in the Figure are not indicated. This makes the kinematics in the Figure difficult to analyze since you have chosen to move the Mollweide projection and not use a fixed projection. You re-used the panel of colors of your Figure 3, but the continents are not in the same color. Please add in the Figure caption the link with the Figure 3 and the motion of the Mollweide projection, which can be confusing for many readers. Since you added the evidence of glaciations on the Figure, do you have any evidence of warm climate to add?

Response: Thanks for this suggestion, we added some text to describe the link of Figure 3 and the motion of the Mollweide projection in the caption of Figure 5. It is the presence and disappearance of glaciations that the study is focused on addressing.

Question 2: Line 14-16 in the supplementary material: the site-mean directions is not calculated by means of Fisher statistics. This is this is a common mistake that is spread in many papers. If you use Bingham statistics (for example) the site-mean directions will be the same. You can-rewrite your sentence like this: “The remanence directions were analyzed using least-squares analysis or great circle intersections as employed by the Paleomagnetism.org web application, site-mean directions were calculated by averaging the different sites, and the statistical parameters were calculated assuming a Fisherian distribution”.

Response: Good suggestion, we have rewritten this paragraph as suggested in the Supplementary Information.

“The remanence directions were analyzed using least-squares analysis¹ or great circle intersections² as employed by the Paleomagnetism.org web application³. Site-mean directions were calculated by averaging the different samples, and the statistical parameters were calculated assuming a Fisher distribution”